# Tracking Equivalent Mechanistic Interpretations Across Neural Networks

**Alan Sun**
Carnegie Mellon University
alansun@andrew.cmu.edu

**Mariya Toneva**
Max Planck Institute for Software Systems
mtoneva@mpi-sws.org

## Abstract

**Mechanistic interpretability (MI)** is an emerging framework for interpreting neural networks. Given a task and model, MI aims to discover a succinct algorithmic process, an **interpretation**, that explains the model's decision process on that task. However, MI is difficult to scale and generalize. This stems in part from two key challenges: there is no precise notion of a valid interpretation; and, generating interpretations is often an ad hoc process. In this paper, we address these challenges by defining and studying the problem of **interpretive equivalence**: determining whether two different models share a common interpretation, *without* requiring an explicit description of what that interpretation is. At the core of our approach, we propose and formalize the principle that two interpretations of a model are equivalent if all of their possible **implementations** are also equivalent. We develop an algorithm to estimate interpretive equivalence and case study its use on Transformer-based models. To analyze our algorithm, we introduce necessary and sufficient conditions for interpretive equivalence based on models' representation similarity. We provide guarantees that simultaneously relate a model's algorithmic interpretations, circuits, and representations. Our framework lays a foundation for the development of more rigorous evaluation methods of MI and automated, generalizable interpretation discovery methods.[1]

## 1 Introduction

Ensuring the interpretability of deep neural networks is central to concerns around AI safety and trustworthiness. Among many proposed interpretation methods, **mechanistic interpretability (MI)** has recently emerged as a promising post-hoc interpretability framework[2] (Olah et al., 2020a; Elhage et al., 2021; Wang et al., 2022, *inter alia*). MI generally operates in two stages: **(a)** identifying a minimal subset of the model's computational graph that drives functional behavior; and **(b)** attaching algorithmic interpretations to each of the recovered mechanisms. In contrast to other attribution-based interpretability methods, MI yields a concrete, human-interpretable algorithmic process that faithfully describes model behavior (Bereska and Gavves, 2024; Koulogeorge et al., 2025). The resulting processes can be used to better understand a model's inductive biases which may inform further improvements (Geva et al., 2021; Meng et al., 2022; Arditi et al., 2024, *inter alia*).

For a fixed task and network, MI approaches can be broadly categorized as either **top-down** or **bottom-up** (Vilas et al., 2024). Top-down methods (b → a) propose a set of high-level candidate algorithms for the task, and then attempt to align those algorithms to the network. While the alignment step can be automated and made statistically rigorous, proposing candidate algorithms is highly ad hoc. For complex tasks, it is often unclear how to even formulate plausible candidates. Additionally, alignment with a proposed algorithm is only a necessary condition for interpretability, and does not guarantee that the model truly implements the intended algorithm (Wu et al., 2023; Geiger et al., 2024; 2025; Sun et al., 2025). In contrast, bottom-up methods (a → b) first isolate mechanisms-of-interest within the model (called a **circuit**) then assign interpretations to components within that circuit (Olah et al., 2020a). Although circuit discovery can be rigorously formulated as

---

[1]Our codebase can be found at https://github.com/alansun17904/interp-equiv

[2]An interpretability method that does not require any retraining. These methods can be applied in parallel with inference and do not compromise the model's performance.

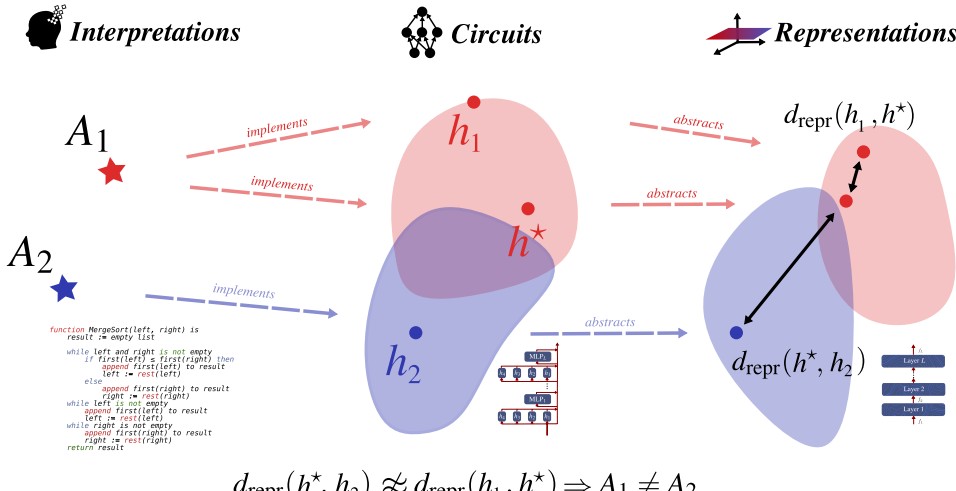

Figure 1: A high-level overview of our algorithmic approach to interpretive equivalence. Consider models $h_1, h_2$ that correspond to possibly *unknown* interpretations $A_1, A_2$ *(Left)*. To determine whether models $h_1$ and $h_2$ are interpretively equivalent, we propose a two-step procedure. First, we sample another model $h^\star$ that also has interpretation $A_1$ *(Center)*. Second, we compare the representation similarity ($d_{\mathrm{repr}}$) between $h_1, h^\star$ and $h^\star, h_2$ *(Right)*. If $A_1, A_2$ are equivalent, then averaged over all implementations $h^\star$, we should not be able to differentiate $d_{\mathrm{repr}}(h_1, h^\star)$ and $d_{\mathrm{repr}}(h^\star, h_2)$.

an optimization problem (Bhaskar et al., 2024), creating and assigning meaningful interpretations to circuits typically demand significant manual effort. Moreover, Méloux et al. (2025) has recently shown that neither top-down nor bottom-up approaches are identifiable: there exists a many-to-many relationship between high-level algorithms and circuits. This ambiguity makes it hard to verify whether a given interpretation is complete and faithful to the model (Jacovi and Goldberg, 2020). We present a detailed discussion of the related literature in Appendix B.

In this paper, we define and study a relaxed, subproblem of MI: **interpretive equivalence**. Concretely, we seek to determine whether two models implement the same high-level algorithm, **without** requiring an explicit description of what that algorithm is. Understanding interpretive equivalence can bridge the gap between bottom-up and top-down approaches through **reductions**. Consider two examples:

**Example 1.1** (Reduction to Simpler Models). MI is computationally prohibitive on larger models (Adolfi et al., 2025). To mitigate this, if we could show that a small model is interpretively equivalent to a large one, then MI analyses on the small model could interpret the larger model.

**Example 1.2** (Reduction to Simpler Tasks). MI's scope is limited because interpreting models that solve complex tasks is at least as hard as manually designing an algorithm to solve them (Nanda et al., 2022; Zhong et al., 2023). Interpretive equivalence provides a criterion to decompose complex tasks into simpler, interpretively equivalent ones.

At the core of our approach (Figure 1), we propose the principle that two high-level algorithms (henceforth termed **interpretations**[3]) are equivalent if their **implementations** are equivalent. Based on this principle, we design an algorithm to detect equivalence by measuring representation similarity. We prove that representation similarity is necessary and sufficient to characterize interpretive equivalence. Overall, our contributions span both theory and practice:

• We propose an algorithm to compute interpretive equivalence between two models *without* interpreting them (Algorithm 1).
• We demonstrate that Algorithm 1 is well-calibrated on a simple task where the ground truth is known. Then, we show its potential to find reductions (Examples 1.1 and 1.2).

---

[3]We distinguish between an algorithm and an interpretation. Informally, an algorithm is independent of any computational abstraction. However, we show in Appendix E that this abstraction-free notion results in pathological equivalence definitions. Thus, we use the terminology **interpretation** to signify a dependence on a chosen level of computational abstraction.

- We specialize the theory of causal abstraction to define interpretations, circuits, and representations (Sections 4 and 5). As byproducts of this formality, we obtain a measure of interpretation quality and an alternative characterization of interpretations as interventions (Section 5 and Appendix H).
- We prove necessary and sufficient conditions for models' interpretive equivalence based on their representation similarity. To our knowledge, these guarantees are the first to relate a model's interpretations, circuits, and representations (Section 6).

Throughout our paper, we use examples from language processing and Transformers. However, our framework does not depend on any properties of the input modality.

## 2  Interpretive Equivalence Through Congruent Representations

We define mechanistic interpretations $A_1, A_2$ as equivalent if they have equivalent implementations. That is, **any model which can be interpreted by $A_1$ must also be interpreted by $A_2$ and vice versa.** We justify this principle later, but first we offer an algorithm to approximate this equivalence. This requires clarifying two procedures: **(1)** how do we enumerate all implementations of $A_1, A_2$? **(2)** How do we measure the distance between these sets of implementations?

**Enumerating Implementations through Interventions.** We call any model $h$ that has a mechanistic interpretation $A$ an **implementation** of $A$ (Definition 5.1). We take a bottom-up approach to generating implementations. Using causal interventions, we identify components in the model that are causally unrelated to the model's behavior (Conmy et al., 2023; Goldowsky-Dill et al., 2023) (i.e. they are not a part of the model's circuit). A model's mechanistic interpretations should be invariant to perturbation or ablation of these components. Dually, each time we perturb or ablate such a component, we yield a "new" model that is causally equivalent to the original one (and as a corollary, shares the same interpretation). Intuitively, every implementation of $A$ should be reachable through a causal intervention applied onto $h$. We take this dual perspective to generating implementations by adding, removing, or modifying these unimportant components (Geiger et al., 2024; Gupta et al., 2024). This procedure is denoted as GETIMPL (Algorithm 1; detailed in Appendix C).

```
1  procedure CONGRUITY(h_1, h_2, n)
2      s ← 0
3      for i ← 1 ... n do
4          h_1, h_1^⋆ ← GETIMPL(h_1)
5          h_2, h_2^⋆ ← GETIMPL(h_2)
6          s ← s + REPRDIST(h_1, h_1^⋆, h_2)
7          s ← s + REPRDIST(h_2, h_2^⋆, h_1)
       return 1 − |s/n − 1|
8  procedure REPRDIST(h_1, h_2, h_3)
9      for i ← 1, 2, 3 do
10         R_i ← GETREPRS(h_i)
11     if d_repr(R_1, R_2) ≤ d_repr(R_1, R_3) then
12         return 1
13     return 0
```

Algorithm 1: Two models are **congruent** if representation similarity alone cannot differentiate them. $d_{\mathrm{repr}}(R_1, R_2)$ measures representation similarity between representations of $h_1, h_2$. GETREPRS, GETIMPL retrieve the hidden representations of a given model and its implementations, respectively.

**Representation Similarity.** We identify each implementation with their hidden representation spaces (GETREPRS in Algorithm 1). Given two implementations, we measure their distance using linear representation similarity ($d_{\mathrm{repr}}$ defined in Definition G.1). Informally, linear representation similarity captures the extent to which the representations of one network can be linearly transformed to the other and vice versa. We notate this as REPRDIST (Algorithm 1).

Suppose that the implementation sets of $A_1, A_2$ were exactly equal. Let $h_1, h_1^⋆$ be implementations sampled from $A_1$ and $h_2$ be an implementation sampled from $A_2$.[4] By symmetry, $\mathbb{P}[d_{\mathrm{repr}}(h_1, h_1^⋆) < d_{\mathrm{repr}}(h_2, h_1^⋆)] = \mathbb{P}[d_{\mathrm{repr}}(h_1, h_1^⋆) > d_{\mathrm{repr}}(h_2, h_1^⋆)]$. In this way, the implementations of $A_1, A_2$ are **congruent** with respect to $d_{\mathrm{repr}}$.

Combining these concepts, we have our approximation of interpretive equivalence: CONGRUITY.

## 3  Experiments

Herein, we demonstrate three applications of Algorithm 1. First, on a toy task where ground-truth interpretations are known, we show that CONGRUITY is well-calibrated (Section 3.1). Next, we apply

---

[4]For the sake of argument, suppose i.i.d. sampling from a uniform distribution over $A_1, A_2$.

our framework to pre-trained language models of various sizes (GPT2 (Radford et al., 2019) and Pythia (Biderman et al., 2023)) and demonstrate that CONGRUITY can distinguish between models that exhibit fine-grained algorithmic differences across various model scales (Section 3.2). Lastly, we show how CONGRUITY can be used to relate a complex task like next-token prediction to a simpler one such as parts-of-speech identification (Section 3.3).

## 3.1 CALIBRATING CONGRUITY

We consider the task of $n$-**Permutation Detection**: determining whether a given sequence of $n$ numbers is a permutation of the elements $1, \ldots, n$. For example, when $n = 3$, [3, 1, 2] $\rightarrow$ True and [1, 2, 2] $\rightarrow$ False. The task is sufficiently rich to support different interpretations, yet simple enough that solutions can be hard-coded as Transformers. We manually devise **six** different interpretations to solve this task whose procedures we detail in Appendix C. Roughly, their approaches can be organized into two buckets:

1. **Sorting-Based (Interpretations 1–4).** First sort the list of numbers by ascending (or descending) order, then directly check whether the resulting sequence is equal to $1, 2, \ldots, n$ (or $n, n-1, \ldots, 1$).
2. **Counting-Based (Interpretations 5–6).** Exploit the fact that the vocabulary contains exactly $n$ numbers and use the pigeonhole principle to detect duplicates in the given sequence.

For all experiments, we fix $n = 10$. Using the Restricted Access Sequence Processing Language (RASP), we hard-code six bidirectional Transformers (of various architectures) to implement each interpretation (Weiss et al., 2021). Then, for each hard-coded RASP Transformer, we leverage a technique of Gupta et al.'s (2024) to generate 100 model variants (with different architectures and weight configurations). These variants are constrained to maintain the same underlying interpretation as their hard-coded counterpart. So for each interpretation, we have 100 implementations. In total, we have $6 \times 100$ models that all achieve 96%+ accuracy on 10-permutation detection. We now directly compute CONGRUITY (Algorithm 1) between pairs of models from our generated implementations. We perform hypothesis testing by bootstrapping a 95% confidence interval over the final output.[5] The results are shown in Figure 2*(Left)*.

Along the diagonals of Figure 2*(Left)*, we see significantly high congruence. So when models share the same interpretation, representations across their implementations are approximately equivalent under linear transformation. On the other hand, off-diagonal entries admit low congruence. Thus, representation similarity across set of implementation as a whole can differentiate individual interpretations. These empirical findings complement our bounds in Main Results 1 and 2 because they suggest that CONGRUITY is both necessary and sufficient to detect interpretive equivalence. Perhaps even more interesting, we observe a breaking row/column in Figure 2, where Interpretations 1–4 (the top-left $4 \times 4$ square) have an average within-group congruence of 0.43. This is larger compared to the average across-group congruence (rows/columns 5-6) with Interpretations 5–6: 0.01. These groupings correspond to our broad algorithmic buckets above and suggest that CONGRUITY could also be adopted as a graded notion to characterize broad-stroke interpretive differences.

## 3.2 REDUCTION TO SIMPLER MODELS

We now consider the task of **indirect-object identification (IOI)** (Wang et al., 2022): given a sentence like "When John and Mary went to the store, John gave the drink to " the model should complete the sentence with "Mary." Because IOI can be solved algorithmically, it has been studied across many models: GPT2-small/medium and the Pythia class of models. While Merullo et al. (2024) find that GPT2-small and medium use the same circuit, Tigges et al. (2024) find that Pythia models across all scales use a consistent but different circuit from GPT2 models (Merullo et al. 2024, Appendix C; Tigges et al. 2024: Appendix D). We use these differences as a practical testbed for CONGRUITY and explore how it generalizes across both model families and scales (85M to 2.9B parameters).[6]

---

[5]Here, $H_0$ : the two models do not share the same interpretation; and our alternate, $H_1$ : the two models share the same interpretations. We compute a Wald confidence interval with 20 bootstrap samples.

[6]Notably, Tigges et al. (2024) also find that circuits in larger models *require more components*, even when the underlying interpretation is constant. Thus, for CONGRUITY to be effective in this setting, it also needs to move beyond identifying architectural similarities.

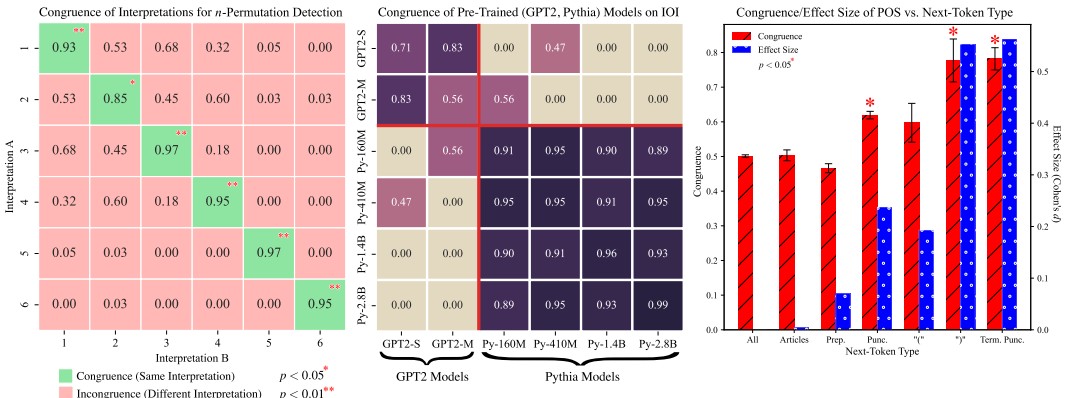

Figure 2: *(Left)* Average congruity between models associated with different interpretations. ■ indicates models have different interpretations; whereas ■ indicates that models have statistically indistinguishable interpretations. *(Center)* Congruity between GPT2 and Pythia family of models on the IOI task. ▬ groups models based on their actual interpretive differences observed by Tigges et al. (2024); Merullo et al. (2024). *(Right)* Congruity between GPT2 on next-token prediction (for different token types: all tokens, articles, prepositions, punctuation, parentheses, and terminal punctuation) vs. GPT2 on in-context parts-of-speech identification.

For each model, we generate 10 parallel implementations by intervening on the components found *not* to be in the IOI circuit (identified by both Tigges et al. 2024 and Merullo et al. 2024). Then, we apply CONGRUITY, each time computing representation similarity with 200 IOI sentences. The results are shown in Figure 2*(Center)*. We find that the Pythia models show high within-group congruence across different scales. GPT2 models exhibit similar behavior. This supports our intuition that interpretive equivalence could be leveraged to reduce the interpretations of complex models into interpretations of smaller ones (Example 1.1). For example, our results affirm that on the IOI task, Pythia-2.8b is interpretively equivalent to Pythia-160M; thus, *a priori*, it suffices to only interpret the latter. Indeed, Tigges et al. (2024) finds that both models share the same interpretation. On average, across-group congruence between Pythia and GPT2 is significantly lower (0.13 versus 0.73 and 0.92) which supports our previous insight that representation similarity provides increased identifiability over interpretations. It is unclear why Pythia-160M and 410M show increased congruence with GPT2-medium and small, respectively; perhaps, subtle similarities in name-mover heads representations could explain this (Tigges et al. 2024, Section 3.2).

## 3.3 REDUCTION TO SIMPLER TASKS

Interpreting next-token prediction poses a challenge for both bottom-up and top-down MI approaches. For top-down approaches writing down a symbolic, end-to-end algorithm for next-token prediction is difficult. Bottom-up approaches face the opposite problem: circuit discovery may identify the entire model as important, failing to reduce the search space of interpretations. We show here that interpretive equivalence may offer some first-steps towards mechanistically understanding next-token prediction. Concretely, we identify sets of next-tokens for which GPT2's next-token prediction process is interpretively similar to parts-of-speech identification (POS[7]).

POS is a largely syntactic task since it focuses solely on the grammatical role of words rather than their meaning in-context. Thus, we expect POS to be interpretively equivalent to the prediction of "syntactic tokens." For computational ease, we conduct all experiments on GPT2. To discover the POS circuit in GPT2, we apply a method of Todd et al.'s (2024). We detail this process, along with the circuit we discover in Appendix C. We construct a dataset for next-token prediction by uniformly sampling 100 sentences from the C4 dataset (Raffel et al., 2020).

We collate tokens from our next-token prediction sentences into disjoint groups of interest. For each group, we apply CONGRUITY and compute representation similarity between our extracted POS circuit

---

[7]Given `tree:noun; run:verb; quick:adverb; fluffy:` the model needs to output `adjective`.

and the last-token hidden representations of the model. The results across different token groups are shown in Figure 2*(Right)*.

As a control, we first consider the group of all tokens. We observe a nonzero congruence of 0.48 with POS. Next, we consider token groups **articles** and **prepositions**. Prediction of these tokens typically appear mid-sentence and depend heavily on semantics.[8] Thus, we expect that predicting these tokens should yield no more interpretive equivalence to POS than the control. Indeed, we find POS's congruity to articles and prepositions to be statistically indistinguishable from the all-token average.

We now heuristically identify two sets of "syntactic tokens:"

1. **Terminal Punctuation** like ".", "?", or "!" mark the end of a sentence. Accurate prediction of these tokens intuitively requires syntactic identification of subject-verb-object relations and (in)dependent clauses.
2. **Closing Brackets/Quotations** like ")" can be implemented as skip-trigrams in one-layer attention Transformers (Elhage et al., 2021). So, we suspect that understanding sentence pragmatics is not needed for their prediction.

We find that both groups yield significantly higher congruence compared our all-token control with medium effect size (measured through Cohen's $d$). This stands in contrast to general punctuation and opening brackets/quotations. Although these other punctuation groups admit statistically significant congruence with POS, we observe a small effect size ($d < 0.3$). We hypothesize that the placement of these tokens rely more on semantic processes. For example, commas may be placed for emphasis or to add dependent clauses rather than strictly maintain grammatical consistency.

## 4 REPRESENTATIONS, CIRCUITS, INTERPRETATIONS AS CAUSAL MODELS

We now present our framework of interpretive equivalence. This theory grounds our algorithmic contributions in Section 2. Whenever possible, we present an informal treatment and defer the precise definitions to Appendix D, where we also have a full glossary and summary of notations. The next sections are organized as follows:

1. In Section 4, we define **circuits**, **representations**, and **interpretations** (Definitions 4.2, 4.3, and 4.4, respectively).
2. In Section 5, we define a criterion for when two interpretations are equivalent. We also introduce **interpretive compression**, a measure of interpretation abstraction.
3. In Section 6, we prove that representation similarity can describe interpretive equivalence (Main Results 1 and 2) and show that CONGRUITY is closely related to these quantities (Main Result 3).

Let $\Sigma$ be an alphabet and $\Sigma^\star$ be all finite strings formed using $\Sigma$. We define a **language model** as a function $h : \Sigma^\star \to \mathbb{R}^{|\Sigma|}$. We often are interested in $h$'s behavior on a subset of inputs $S \subset \Sigma^\star$, rather than on all of $\Sigma^\star$. For this reason, we term $S$ a **task**. While $h$ alone serves as a sufficient functional description, interpretability requires understanding $h$'s underlying computational process. For this, we rely on three behavioral characterizations of a neural network at different levels of abstraction: $h$'s **circuits**, **representations**, and **interpretation**.

### 4.1 CIRCUITS

**Definition 4.1** (Deterministic Causal Model, Geiger et al. 2025). A (deterministic) causal model with $m$ components is a quadruple $(\mathbb{V}, U, \mathbb{F}, \succ)$ where $\mathbb{V} = (\boldsymbol{v}_1, \ldots, \boldsymbol{v}_m)$ is a set of **hidden** variables such that $|\mathbb{V}| = |\mathbb{F}| = m$, $U$ is an **input** variable, and $\succ$ defines a partial ordering over $\mathbb{V}$. For each $\boldsymbol{v}_k \in \mathbb{V}$,

$$\boldsymbol{v}_k \triangleq f_k(Pa(\boldsymbol{v}_k), U) \tag{4.1}$$

where $f_k \in \mathbb{F}$ and $Pa(\boldsymbol{v}_k) \subset \{\boldsymbol{v} \in \mathbb{V} : \boldsymbol{v} \succ \boldsymbol{v}_k\}$ are the **parents** of $\boldsymbol{v}_k$.

---

[8]Consider the fragment: "`[context]` The students are looking `[mask]`" the next prepositions—"at", "into", or "for"—are all syntactically sound, but they are semantically ambiguous without further context. Articles are similar: consider, "I need to buy `[mask]` car." `[mask]` could be either the indefinite ("a") or definite ("the"), both are syntactically valid but the choice requires pragmatic reasoning.

A deterministic causal model describes a computational graph where hidden variables $\mathbb{V}$ (nodes) store latent computation results computed by the functions in $\mathbb{F}$ (edges). Since a computation graph is acyclic, the data/compute flow determined by $\mathbb{F}$ induces a partial ordering over $\mathbb{V}$, $\succ$.

For a specific input setting $U \leftarrow u$, we denote by $\mathbb{V}^{\mathbb{F}}(u)$ to be the unique **solution**: a configuration of the values $\mathbb{V}$ consistent with both $u$ and $\mathbb{F}$ (Peters et al., 2017). Likewise, $v_i^{\mathbb{F}}(u)$ is the solution of $v_i \in \mathbb{V}$ to $U \leftarrow u$. **In this way, $v_i^{\mathbb{F}}$ is a function that maps from $S \to \mathbb{R}^{n_i}$.**

There is a natural tradeoff between the collective complexity of $\mathbb{V}$ and the complexity of any individual operator in $\mathbb{F}$. On one extreme, any blackbox model $h$ can be expressed as a **trivial causal model** with one hidden variable: $v_1 \triangleq h(U)$. On the other, each $v \in \mathbb{V}$ could be a single neuron (thus $|\mathbb{V}| \sim 10^9$), while elements in $\mathbb{F}$ consist of primitive operations like dot-products. In this way, $\mathbb{V}$ determines the level of abstraction of the causal model. We now use this formalism to define circuits.

**Circuits** form the starting point for almost all bottom-up approaches in MI (Wang et al., 2022; Conmy et al., 2023; Lieberum et al., 2023, *inter alia*). Intuitively, for some task $S$, a circuit describes the (minimal) computational pathways used by the model to produce $h(S)$. Throughout our treatment, we do not require minimality as finding minimal circuits has been shown to be an intractable combinatorial optimization problem (Adolfi et al., 2025).

**Definition 4.2** (Circuit). An $m$-circuit of $h$ on $S$ is a causal graph $(\mathbb{V}, U, \mathbb{F}, \succ)$ with $m$ components that satisfies: **(1)** $U$ is $S$-valued; **(2)** Hidden variables are real-valued; **(3)** There exists a path from $U$ to a variable $v_{\text{out}} \in \mathbb{V}$ such that $v_{\text{out}}^{\mathbb{F}}(S) = h(S)$.

For any $h$ and $m \in \mathbb{Z}^+$, an $m$-circuit of $h$ exists by its trivial causal model. Since we do not require minimality, $m$-circuits are not unique: a joint change of bases to any $v_k \in \mathbb{V} \setminus v_{\text{out}}$ and $f_k$; or, adding a new variable that does not contribute to $v_{\text{out}}$ results in a new causal model that preserves faithfulness to $h$. We further explore some of these properties in Section 5.

## 4.2 Representations

**Representations** of a neural network are a sequence of activation spaces. Increasingly, representations have been understood to encode concepts which deep networks then iteratively refine (Jastrzebski et al., 2018). We formally view representations as abstractions of circuits and define them through **causal abstraction** (Beckers and Halpern, 2019; Geiger et al., 2025). A causal model $K^\star$ **abstracts** $K$ if there exists a surjective map from $K$'s variables to $K^\star$'s variables that preserves $K$'s causal relationships. Any map that admits this abstraction is an **alignment** from $K$ to $K^\star$ (Definition D.3). Essentially, an alignment $\pi$ maps $K$'s "low-level" variables to $K^\star$'s "high-level" variables (Rubenstein et al., 2017). Like $m$-circuits, for a given low-level and high-level causal model pair, the set of alignment maps that exist between them are not unique.

**Definition 4.3** (Representations). Let $K$ be an $m$-circuit of $h$ on a task $S$. An $L$-circuit that abstracts $K$ is an $(L, \delta)$-representation of $K$ if for each $v_k \in \mathbb{V}$,

- $Pa(v_{k+1}) = \{v_k\}$, $Pa(v_1) = \{U\}$, and $v_L = v_{\text{out}}$.
- For each $1 \leq k \leq L$, there exists linear maps $A_k : \mathbb{R}^{n_k} \to \mathbb{R}^{|\Sigma|}, B_k : \mathbb{R}^{|\Sigma|} \to \mathbb{R}^{n_k}$ where $\|A_k\|_{\text{op}}, \|B_k\|_{\text{op}} \leq 1$,

$$\|A_k v_k^{\mathbb{F}} - h\| < \delta \qquad \text{and} \qquad \|B_k h - v_k^{\mathbb{F}}\| < \delta \qquad (4.2)$$

for some norm (to be specified).

Representations have two crucial properties: they compress the complex causal model of $K$ into a single chain of length $L$; and, they have high signal and low noise. By Equation (4.2), each layer's hidden representation can identify $K$'s output behavior (independent of its children). Yet, the representations at each layer are simple enough that outputs of $h$ can identify their value (independent of its ancestors). In MI, this latter constraint is often used as a search criterion to localize a network's circuit (Belrose et al., 2025). For $h$, its set possible of circuits explode combinatorially with depth, whereas the set of possible representations only scale quadratically. Therefore, compression through abstract representation gives us a tractable staging ground to compare circuits.

### 4.3 Interpretations

**Interpretations** are symbolic, human-understandable descriptions of $h$ that faithfully captures its functional behavior on a task. We formalize them as abstractions.

**Definition 4.4.** Fix a task $S$. A causal model $A \triangleq (\mathbb{V}^\star, U, \mathbb{F}^\star, \succ)$ that abstracts a circuit $K \triangleq (\mathbb{V}, U, \mathbb{F}, \succ)$ is an $\eta$-faithful interpretation if $A$'s output approximates $K$'s with error at most $\eta$ across inputs in $S$.

Notably, our construction does not assume there is a distance metric over $A$'s variables nor do we specify the values of these variables. This generality aligns our framework with most MI literature where $\mathbb{V}^\star$ could take on symbolic values (Hanna et al., 2023; Zhong et al., 2023, *inter alia*).

For any circuit $K$, an interpretation always exists through the **trivial interpretation**: $K$ itself is an interpretation of $K$ (with the identity abstraction; albeit, this is not very useful). Since interpretations depend on a fixed level of abstraction ($\mathbb{V}^\star$), they must *not* be unique: by varying $\mathbb{V}^\star$ and $\pi$ we can produce different interpretations.[9] As a corollary, circuits are *not* identifiable by their interpretations. This seems problematic, but in Appendix E we demonstrate that abstraction-dependent interpretations are necessary for a non-trivial notion of interpretive equivalence.

## 5 Interpretive Equivalence through Implementation Equivalence

Méloux et al. (2025) shows that there exists a many-to-many relationship between circuits and interpretations. This implies that comparing individual circuits and interpretations is ill-defined. Thus, we propose that two interpretations are (approximately) equivalent if and only if their implementations are (approximately) equivalent. By examining families of circuits instead of individual ones, we can effectively quotient out the many-to-one mapping from circuits to interpretations.

To start, we define **implementations**. Let $K \triangleq (\mathbb{V}, U, \mathbb{F}, \succ)$ be an $m$-circuit of $h$ on some task $S$. Let $A \triangleq (\mathbb{V}^\star, U, \mathbb{F}^\star, \succ)$ be an $\eta$-faithful interpretation of $K$ through an alignment $\pi : \text{SubsetsOf}(\mathbb{V}) \to \mathbb{V}^\star$ (see Definition D.3). Notice that for fixed $A, \pi$, by carefully varying $\mathbb{F}$ we could construct many different circuits (and computation graphs) over $(\mathbb{V}, U)$ that all abstract to the same interpretation $A$ under $\pi$. **Concretely, in our framework, $\mathbb{V}, U$ defines a neural architecture and $\mathbb{F}$ corresponds to many weight configurations that result in the same interpretation $A$ under $\pi$.** We then define $A$'s implementations as **the set of all admissible $\mathbb{F}$ under $A, \pi$.** This construction mirrors observations that implementations are diffuse and plentiful in the weight space (Lubana et al., 2023; Zhong et al., 2023; Gupta et al., 2024). Precisely,

**Definition 5.1** (Implementation). Let $\Pi$ be a class of alignments. $F$ is an implementation of $A$ if there exists $\pi \in \Pi$ such that $A$ is an $\eta$-faithful interpretation of $(\mathbb{V}, U, F, \prec_F)$ under $\pi$.

By a slight abuse of notation, we denote $\Pi^{-1}(A)$ as the set of all implementations of $A$ under alignments $\Pi$. Then, a non-empty intersection between $\Pi_1^{-1}$ and $\Pi_2^{-1}$ directly corresponds to Méloux et al.'s (2025) non-identifiability result. We assume that any circuit comes with a metric attached to its variables, i.e. $d : \mathbb{V} \times \mathbb{V} \to \mathbb{R}^{\geq 0}$. This is reasonable, as a circuit's variables are real-valued (Definition 4.2). $d$ naturally induces a pseudometric over $\Pi^{-1}(A)$, where for $F, \tilde{F} \in \Pi^{-1}(A)$

$$d(F, \tilde{F}) \triangleq d(\mathbb{V}^F, \mathbb{V}^{\tilde{F}}) = d(\mathbb{V}^F(S), \mathbb{V}^{\tilde{F}}(S)). \quad (5.1)$$

Given two sets of implementations over $(\mathbb{V}, U, \cdot, \cdot)$ we can leverage $d$ to quantify (1) the Hausdorff distance between them; and, (2) their diameters. We call the former **interpretive equivalence** and the latter **interpretive compression** (see Figure 3). More precisely,

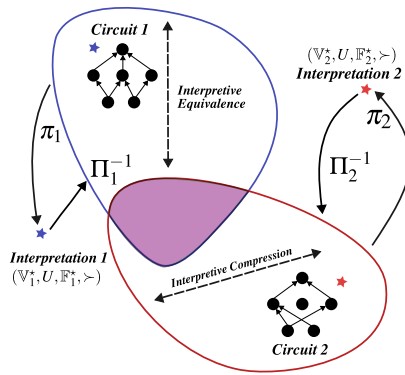

Figure 3

**Definition 5.2.** For $i \in \{1, 2\}$, let $A_i \triangleq (\mathbb{V}_i^\star, U, \cdot, \cdot)$ be a $\eta_i$-faithful interpretation of $K \triangleq (\mathbb{V}, U, \cdot, \cdot)$ and $\Pi_i$ be a class of alignments that map from $\text{SubsetsOf}(\mathbb{V}) \to \mathbb{V}_i^\star$. Then, $A_1, A_2$ are $\varepsilon$-**approximate interpretively equivalent** if

$$d_{\text{interp}}(A_1, A_2) \triangleq d_{\text{H}}\left(\Pi_1^{-1}(A_1), \Pi_2^{-1}(A_2)\right) < \varepsilon, \quad (5.2)$$

---

[9]To see more details of this construction, refer to Definition D.3 and Equation (D.2)

where $d_{\mathrm{H}}$ is the Hausdorff distance[10] defined by $d$ (see Equation (5.1)).

**Definition 5.3.** Let $A$ be an $\eta$-faithful interpretation of $K$ and $\Pi$ be a class of alignments. Then, the **interpretive compression** of $A$ is

$$\kappa(A, K, \Pi) \triangleq \mathrm{diameter}(\Pi^{-1}(A)) = \sup_{F, F' \in \Pi^{-1}(A)} d(F, F').  \tag{5.3}$$

Interpretive equivalence and compression are tightly coupled. To explore their relationship, let us consider two extremes. On one hand, suppose $A = K$ and $|\Pi| = 1$. Then, there exists exactly one implementation of $A$ under $\Pi$: $K$. This implies that there is no interpretive compression. Further, any interpretation equivalent to $A$ must admit the same exact same weight configuration as $K$. This effectively creates a bijective mapping between model weights and interpretations. On the other hand, suppose that $A$ is the trivial causal model of $h$, and let $\Pi$ be all alignments. Since $A$ conveys no information about the computational process of $K$ and $\Pi$ places no constraint on alignment, any circuit by Definition 4.2 must be an implementation of $A$. In fact, Sutter et al. (2025) show under mild assumptions that this phenomenon persists as long as $\Pi$ is the set of all alignments. And so, in this case, interpretive compression is maximal. These examples illustrate a duality between equivalence and compression: while more compression shrinks and simplifies the set of possible interpretations, it simultaneously enlarges the set of implementations and complicates interpretive equivalence.

Interpretive equivalence, in Definition 5.2, requires that both interpretations originate from the same circuit $K$. This constraint can be relaxed by augmenting $\Pi_i$ such that $(\pi : \mathrm{SubsetsOf}(\mathbb{V}_1) \to \mathbb{V}_i^\star) \in \Pi_i$. Through $\Pi_i^{-1}$, we only consider implementations of $A_1, A_2$ on the same neural architecture $\mathbb{V}_1$. This creates asymmetry in choosing which circuit to use for comparison ($\mathbb{V}_1$ or $\mathbb{V}_2$). Our intuition is that we should select the circuit of least compression that admits at least one implementation for both interpretations.[11] Our bounds in Section 6 justify this principle.

We describe critical properties of interpretive equivalence and compression in Appendix F.

## 6 Representational Similarity and Interpretive Equivalence

Using interpretive compression and representation similarity, we now prove both upper and lower bounds on interpretive equivalence (Main Result 1 and Main Result 2, respectively). The proofs and formalisms are deferred to Appendix G. Throughout this section, we consider the most general setup: for $i \in \{1, 2\}$, let $K_i$ be a circuit of a model $h_i$ over a shared task $S$. Assume each circuit admits an $(L_i, \delta_i)$-representation $R_i$ through an alignment map $\rho_i$. First, we define a notion of distance between these two representations.

**Definition 6.1** (Representation Similarity)**.** Let $H_i$ be the concatenation of all hidden variables in $R_i$. Then, the representation similarity of $R_1, R_2$, $d_{\mathrm{repr}}(R_1, R_2)$, is the bidirectional linear approximation error between $H_1, H_2$ (formally, Definition G.1).

Defining similarity with a strict linear approximation is not necessary to develop our theory, but offers attractive computational tradeoffs. This is because we can formulate representation similarity as multi-target linear regression for which there are sharp approximations with small sample complexity. Alternatively, $d_{\mathrm{repr}}$ could be defined with an arbitrary kernel (Kornblith et al., 2019). We also make the following assumptions: **(1)** $\rho_i$ is Lipschitz continuous with Lipschitz constant $\mathrm{Lip}(\rho_i)$. **(2)** There exists $\Delta > 0$ such that $\|h_1 - h_2\| < \Delta$ over $S$. **(3)** The implementation distance $d$ (see Equation (5.1)) is induced by a fixed **distillation** function $\Phi$: for implementations $F, \tilde{F}$ over the same variables $\mathbb{V}$, $d(F, \tilde{F}) \triangleq \|\Phi\mathbb{V}^F - \Phi\mathbb{V}^{\tilde{F}}\|$. **(4)** There is a 1-Lipschitz map $\Psi$ such that $\|\Psi H_1 - \Psi H_2\| \le d_{\mathrm{repr}}(R_1, R_2)$ and $\|\Phi\mathbb{V}^{\mathbb{F}} - \Psi H\| < \omega$.

Assumption 1 stabilizes representations such that a small change in circuitry implies a proportional representation perturbation. In practice, this is reasonable since representations are usually distilled from neural networks via linear projection or by averaging the activations of several neural components (Sucholutsky et al., 2023). Assumption 2 guarantees that functional differences between our

---

[10]The Hausdorff distance between two sets $A, B$ is the greatest distance a point from $A, B$ needs to travel to reach $B, A$, respectively: $\tilde{d}(a, B) \triangleq \inf_{b \in B} d(a, b)$, $d_{\mathrm{H}}(A, B) \triangleq \max(\sup_{a \in A} \tilde{d}(a, B), \sup_{b \in B} \tilde{(b, A)})$.

[11]Attached to each $\mathbb{V}_1, \mathbb{V}_2$ are distance metrics $d_1, d_2$. Alternatively, one could also choose the circuit where $d_i$ is most convenient to compute.

target models are bounded, otherwise their interpretive *inequivalence* is trivial. Assumptions 3-4 state that our implementation metric $d$ only measures distance using a set of observable components. Further, these components are recoverable from representations (up to $\omega$). In this way, representations can serve as a surrogate for distances between circuits.

For $i \in \{1,2\}$, suppose that $A_i$ is an $\eta_i$-faithful interpretation of $K_i$. We first show that representation similarity between $K_1, K_2$ upperbounds interpretive equivalence.

**Main Result 1.** *Let $\Pi$ and $\Pi_\star$ be alignment classes that map $K_1$'s variables into $A_1, A_2$'s variables, respectively. Suppose there exists $F^\star \in \Pi_\star^{-1}(A_2)$ such that its representation under $\rho_1$ is an $(L_1, \delta_1)$-representation. Then, the approximate interpretive equivalence of $A_1, A_2$ under alignments $\Pi^{-1}, \Pi_\star^{-1}$ is*

$$d_{\text{interp}}(A_1, A_2) \leq \underbrace{\kappa(A_1, K_1, \Pi) + \kappa(A_2, K_1, \Pi_\star)}_{\text{(a)}} + \underbrace{2\omega + d_{\text{repr}}(R_1, R_2) + \delta_1 + \delta_2 + \Delta}_{\text{(b)}}. \tag{6.1}$$

Equation (6.1) explicits two dependencies: **(a)** measures interpretive compression of $A_1, A_2$ on $K_1$. When compression is low, circuits retain more information about interpretive differences. And since representations distill circuits, it is intuitive that representation similarity dominates the upperbound in this regime. In contrast, when compression is high, our examples in Section 5 illustrate that when circuits become uninformative, so should representations and their similarities; **(b)** jointly measures the quality of representations $R_1, R_2$ and their differences. If $\delta_1, \delta_2 \gg 1$, representations lose meaningful linear structure even though they sufficiently abstract implementations of $A_1, A_2$. This bound exposes a tension between the geometric richness of representations and our computational approach to their similarities. For example, representations lying on complex manifolds may not be well-described by our linear formulation of representation similarity (Ansuini et al., 2019; Pimentel et al., 2020). Thus, we hypothesize representations and their similarity criterion need to be simultaneously broadened to achieve more generality. We leave the implications of this for future work. Now, the lowerbound:

**Main Result 2.** *Let $\Pi$ and $\Pi_\star$ be an alignment that map $K_1$'s variables onto $A_1, A_2$'s variables. Suppose $A_1, A_2$ are $\varepsilon$-approximate interpretively equivalent under the alignment classes $\Pi_\star, \Pi$. Then,*

$$\varepsilon \geq \frac{d_{\text{repr}}(R_1, R_2) - (\delta_1 + \delta_2 + \Delta)}{\text{Lip}(\rho_1) + 1}. \tag{6.2}$$

The terms in Equation (6.2) mirror the dynamics of Equation (6.1). If the implementation sets are $\varepsilon$-close, then there cannot be too much discrepancy between the observed representations (see the numerator), unless it can be explained away by the slack terms $(\delta_1 + \delta_2 + \Delta)$. Along these lines, $\text{Lip}(\rho_1)$ (in the denominator) captures the intuition that when the mapping from circuits to representations is unstable, small implementation-level gaps can lead to larger representational differences. Beyond interpretive equivalence, Main Result 2 has implications for steering (Singh et al., 2024). When the identifiability of representations is low ($\text{Lip}(\rho_1) \gg 1$ and $d_{\text{repr}}(R_1, R_2) \ll 1$), steering interventions are less likely to induce interpretive change, i.e. significant alterations to the representation space may see little change in model interpretation. In Main Result 3, we show how this directly relates to CONGRUITY. For $i \in \{1,2\}$, let us assume an arbitrary distribution over $\Pi_i^{-1}$ and denote $\mathbb{P} \triangleq \mathbb{P}_1 \otimes \mathbb{P}_1 \otimes \mathbb{P}_2 \otimes \mathbb{P}_2$ the product distribution.

**Main Result 3.** *Let $p$ be the expected value of REPRDIST in Algorithm 1, $F_1, F_2 \overset{iid}{\sim} \Pi_1^{-1}(A_1)$, and $F_3 \sim \Pi_2^{-1}(A_2)$ then*

$$p \geq 1 - \inf_{\varepsilon > 0} \mathbb{P}[d_{\text{repr}}(F_1, F_2) > \kappa(A_1, K_1, \Pi_1) + \varepsilon] - \mathbb{P}[d_{\text{repr}}(F_1, F_3) < d_{\text{interp}}(A_1, A_2) - \varepsilon]. \tag{6.3}$$

This lowerbound demonstrates that REPRDIST (and as a corollary, CONGRUITY; see Corollary H.1) accounts for interpretive compression and equivalence. Concretely, it predicts that, in expectation, REPRDIST will be large when $\Pi_1^{-1} \neq \Pi_2^{-1}$ except in two cases: **(a)** within-interpretation dispersion is high: the representations of two implementations of $A_1$ can differ more than their interpretive compression; or **(b)** cross-interpretation collision: an implementation of $A_2$ lies close in representation to $A_1$ despite $d_{\text{interp}}(A_1, A_2) \gg 1$. We discuss these results further in Appendix H and also analyze the sample complexity of CONGRUITY.

## 7 CONCLUSION

In this paper, we define and study the problem of interpretive equivalence: detecting whether two models share the same mechanistic interpretation without interpreting them. We contribute an algorithm to estimate equivalence and demonstrate its use on toy and pre-trained language models. Then, we provide a theoretical foundation to ground and explain our algorithm. Our framework and results lay a foundation for the development of more rigorous evaluation methods in MI and automated, generalizable interpretation discovery methods.

## ACKNOWLEDGEMENTS

Alan is supported by the National Science Foundation Graduate Research Fellowship Program under Grant No. DGE2140739. Alan thanks Büşra Asan and Andrew Koulogeorge for helpful feedback and discussion of early drafts of the work.

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

## A  QUESTIONS AND CLARIFICATIONS

> *It seems that a prerequisite of CONGRUITY is circuit discovery. Given that circuit discovery is NP-HARD, how can this algorithm be efficient?*

*A:* A **circuit** is a minimal subset of the computation graph that explains the model's functional behavior. Notably, this minimality requirement is what makes circuit discovery hard. In CONGRUITY, we do not require a circuit in a strict sense. Instead, the practitioner needs to identify **individual** components that are **unimportant**, i.e. removing it does not affect functional behavior. This is more relaxed than performing circuit discovery and applying the complement. For a model with $m$ components, existing component scoring methods such as activation/attribution patching can effectively identify unimportant component sets in $O(1)$ time (Conmy et al., 2023; Nanda, 2023).

> *In Section 5, it is argued that there could be many interpretations for a given circuit. How do we know which (or at what level) interpretations are being compared through CONGRUITY, if we do not interpret the models first?*

*A:* The level at which interpretations are being compared in CONGRUITY depends wholly on the interventions being performed in GETIMPL. Given an interpretation $A$ of some circuit $K$, consider the set of $A$'s implementations under some arbitrary alignment class $\Pi$. We could try and enumerate the set $\Pi^{-1}(A)$ naively. In most cases, this is extremely difficult because $\Pi^{-1}(A)$ could have complex geometry in the weight space (Lubana et al., 2023). We could identify this set by dually enumerating interventions. Concretely, fix some $F \in \Pi^{-1}(A)$. Construct a set of interventions $\Gamma$ such that for each $\tilde{F} \in \Pi^{-1}(A)$, there is $\gamma \in \Gamma$ where $\gamma(F) = \tilde{F}$. In many cases, $\Gamma$ is more intuitive than $\Pi^{-1}(A)$ (Geiger et al., 2024; Park et al., 2024; Gupta et al., 2024). Alternatively, we could directly define $\Pi^{-1}(A)$ with $\Gamma$. This strategy is employed in GETIMPLto implicitly define the set of interpretations being compared. To our knowledge, there is surprisingly little literature on the nature of these interventions and we believe that our insights in Sections 2 and 5 suggest this may be a fruitful direction for future work.

> *The tasks presented in Section 3 are quite simple. Would linear representation alignment as a similarity metric between representations in CONGRUITY still be effective for more complex tasks?*

*A:* We choose linear representation alignment because of the linear representation hypothesis (Park et al., 2024). However, this does not restrict the generality of our framework. This is because the choice of alignment map from circuits to representations, $\rho$, is unrestricted. One could indirectly construct more complex $d_{\text{repr}}$ by indirectly increasing the complexity of $\rho$. In this case, $\rho$ can be thought of as a constant feature map of arbitrary complexity.

> *How does the proposed framework allow practitioners to translate interpretive equivalence into actual interpretations?*

*A:* Interpretive equivalence alone is not sufficient to identify interpretations. We leave the formal exploration of this for future work. But, as discussed in Examples 1.1 and 1.2, interpretive equivalence potentially simplifies the interpretation workflow.

> *How would the theoretical framework in Sections 4 and 5 generalize to different notions of causal abstraction?*

*A:* The difference between various causal abstraction frameworks is their definition of an alignment map $\pi$ (see Definition D.3). For example, **bijective translations** (in contrast to **constructive abstractions**, the notion of abstraction we use throughout the main text) require that the alignment map be bijective. Our framework can easily accommodate these different notions by simply tightening or loosening the set of permissible alignments $\Pi$. Further, assuming all else equal, as $\Pi$ increases or decreases in size, interpretive compression necessarily increases and decreases, respectively. In this way, the tightness of Main Results 1 and 2 adjusts to the chosen notion of causal abstraction. On the other hand, Main Results 1 and 2 do not rely on structural properties of $\Pi$. Potential future work could significantly tighten these bounds by considering sharp restrictions over the allowed alignments.

> *Why is it that in Main Results 1 and 2, there is no dependence on the $\eta_i$-faithfulness of the interpretations that we are comparing?*

*A:* Faithfulness is implicitly encoded into interpretive compression which appears in both Main Results 1 and 2. Let $A$ be an $\eta$-faithful interpretation of $K$. Fix some alignment class $\pi$ and consider $\Pi^{-1}(A)$. As $\eta \to \infty$, the size of $\Pi^{-1}(A)$ grows monotonically. To see this, let $\varepsilon > 0$ be arbitrary. If $A$ is an $\eta$-interpretation of $K$, then $A$ must also be an $(\eta + \varepsilon)$-interpretation of $K$. In other words, if the faithfulness constraint is relaxed, then $A$ could be a faithful interpretation for any circuit. This implies that any notion of interpretive equivalence must be trivial since every interpretation corresponds to every circuit.

## B   RELATED LITERATURE

There has been extensive study in interpreting neural networks (specifically, language models). With respect to our paper, they can be organized into the following three buckets.

**Mechanistic Interpretability.** Mechanistic Interpretability (MI) can be broadly broken into two distinct phases: first, identifying a minimal subset of the model's computational graph that is responsible for a specific behavior—this process is also referred to as *circuit discovery* (Olah et al.,

2020b;a; Conmy et al., 2023; Bhaskar et al., 2024; Hanna et al., 2024); second, assigning human-interpretable explanations to each of the extracted components—this process is generally referred to as *mechanistic interpretability* (Chan et al., 2022; Wang et al., 2022; Zhong et al., 2023; Hanna et al., 2023; Merullo et al., 2024). This paper focuses on the latter process; thus, when we invoke the term mechanistic interpretability, we are referring specifically to this second phase. For completeness, we briefly review circuit discovery as well.

A model can be seen as a computational graph (Elhage et al., 2021; Olsson et al., 2022): $G = (V, E)$ with vertices $V$ and edges $E \subset V \times V$. A circuit is then any binary function $f : E \to \{0, 1\}$. The optimal circuit $f$ satisfies two properties: (1) it minimizes $\sum_{e \in E} f(e)$; and (2) when edges not in the circuit $e \in E, f(e) = 0$ are ablated, the model's functional behavior remains unchanged. Although circuit discovery can be concretely formulated as an optimization problem, it is computationally intractable in its naive form (Bhaskar et al., 2024; Adolfi et al., 2025). As a result, many relaxations and heuristics have been developed (Conmy et al., 2023; Syed et al., 2023; Nanda, 2023; Hanna et al., 2024; Bhaskar et al., 2024). Nevertheless, there exists a solution set that can be statistically verified (Shi et al., 2024). The process of mechanistic interpretability, which we focus on in this paper, presents different challenges. MI involves iteratively generating hypotheses about the interpretations for the interactions between different components, then testing those hypotheses through carefully crafted interventions (Chan et al., 2022; Geiger et al., 2024; Méloux et al., 2025; Sun et al., 2025, *inter alia*). MI methods can be further clustered into two approaches: *top-down* and *bottom-up*. Top-down approaches start by enumerating hypotheses about the possible algorithms the model could be implementing. Then, they iterate through these hypotheses and isolate those with the closest causal alignment (Wu et al., 2023; Geiger et al., 2024; Bereska and Gavves, 2024; Vilas et al., 2024; Sun et al., 2025). On the other hand, bottom-up approaches are data-driven. They seek to directly find algorithmic explanations of the model's mechanistic behaviors by analyzing activations or attention patterns (Nanda et al., 2022; Zhong et al., 2023; Lee et al., 2024; Arditi et al., 2024; Nikankin et al., 2025).

**Casual Abstraction.** Causal abstraction seeks to formally characterize when a symbolic explanation faithfully explains a data-generating process (in our application, this data-generating process would be a model) (Pearl, 2009; Peters et al., 2017; Beckers and Halpern, 2019; Beckers et al., 2020; Otsuka and Saigo, 2022). Causal abstraction as defined in Definition D.3 seeks to align "low-level" models (neural networks) with "high-level" models (symbolic explanations). These theoretical constructions have driven the development of many top-down mechanistic interpretability methods such as Wu et al. (2023); Geiger et al. (2024); Sun et al. (2025) which directly verify this condition. While causal abstraction provides a necessary condition for a valid interpretation, recent works such as Méloux et al. (2025) argue that it alone is insufficient to fully characterize what constitutes a valid interpretation. Practical procedures that stem from this theory also fail to address this. This highlights the need for additional theoretical frameworks to complement causal abstraction.

On the other hand, causal abstraction represents a *hard* equivalence: two models either *are* or *are not* causal abstractions of each other. The binary nature of this definition fails to capture our intuition that explanations can vary in quality or completeness. Some interpretations may better capture the model's behavior than others, or may only explain a subset of the model's functionality. The hard equivalence of causal abstraction makes it difficult to reason about these partial or imperfect explanations. As a result, framemworks such as Beckers et al. (2020); Massidda et al. (2023) soften this criterion by introducing distance metrics on the total settings of the "high-level" model. In practice, these approaches face two main drawbacks:

1. It is challenging to define meaningful distance metrics when high-level interpretations involve discrete symbols and symbolic reasoning.
2. Comparing interpretations between models requires fully interpreting each model first, which forces a computationally expensive top-down analysis approach.

Our framework addresses these limitations by focusing on *implementations* rather than interpretations directly. Since neural network computations are fundamentally real-valued, we can leverage natural distance metrics in their computational space. Additionally, by analyzing families of implementations rather than requiring complete interpretations, we can compare models' interpretive similarity without the overhead of fully interpreting each model first.

**Representation Similarity.** Representation similarity in neural networks has emerged as a fundamental research area whcih addresses how internal representations correspond across different models, domains, and biological systems (Kornblith et al., 2019). We provide a brief overview of techniques to measure representation similarity, but encourage the reader to refer to Sucholutsky et al. (2023); Klabunde et al. (2025) for a comprehensive survey of the techniques within this field.

Representations of deep neural networks have shown to exhibit an array of powerful properties. They have given insights into training dynamics and the generalization capabilities of models (Tishby et al., 2000; Xu and Raginsky, 2017; Shwartz Ziv and LeCun, 2024) and also provide a tractable method to compare the inductive biases between different models (Wang and Isola, 2020; Skean et al., 2025). These representations have also shown to admit rich geometric properties (Tulchinskii et al., 2023). In this paper, we measure representation through a learned linear regression. These techniques have been used extensively to compare different neural architecture and even human-language model similarity (Toneva and Wehbe, 2019; Muttenthaler et al., 2023).

## C  EXPERIMENTAL DETAILS

In this section, we detail our experimental methods. We first review constructs of RASP Weiss et al. (2021). Then, we demonstrate that RASP provides an interface to craft interpretations, and through methods like Geiger et al. (2024); Gupta et al. (2024) we can enumerate implementations of these interpretations. Lastly, we describe our experimental hyperparameters.

### C.1  RASP INTERPRETATIONS

**Background on RASP Programs.** The Restricted Access Sequence Programming (RASP) language is a functional programming model designed to capture the computational behavior of Transformer architdectures (Weiss et al., 2021). RASP programs have shown use in mechanistic interpretability both as an effective benchmarking tool for faithfulness (Conmy et al., 2023; Hanna et al., 2024) and as a method to develop "inherently" interpretable language models (Friedman et al., 2023). Another line of work uses it (and other similar methods) as a proof technique to reason about the Transformer architecture's generalizability on a host of tasks (Weiss et al., 2021; Merrill et al., 2022; Giannou et al., 2023). In this paper, we focus on RASP's applications in interpretability.

RASP programs operate on two primary types of variables: *s-ops*, representing the input sequence, and *selectors*, corresponding to attention matrices. These variables are manipulated through two fundamental instructions: elementwise operations and select-aggregate. *Elementwise operations* simulate computations performed by a multilayer perceptron (MLP), while *select-aggregate* combines token-level operations, modeling the functionality of attention heads.

Every RASP program is equipped with two global variables `tokens` and `indices`, essentially primitive *s-ops*. `tokens` maps strings into their token representations:

```
token("code") = ["c", "o", "d", "e"]
indices("code") = [0, 1, 2, 3]
```

On the other hand, `indices` map $n$-length strings into their indices. That is, a list of $[0, 1, \ldots, n-1]$. Elementwise operations can be computed through composition. That is,

```
(3 * indices)("code") = [0, 3, 6, 9]
(sin(indices)) = [sin(0), sin(1), sin(2), sin(3)]
```

Tokens and their indices can also be mixed through *selection matrices* which are represented through the *s-op select*. This operations captures the mechanism of the QK-matrix. It takes as input two sequences $K, Q$, representing keys and queries respectively, and a Boolean predicate $p$ and returns a matrix $S$ of size $|K| \times |Q|$ such that $S_{ij} = p(K_j, Q_i)$. Then, the OV-circuit can be computed through the *select-aggregate* operation, which performs an averaging over an arbitrary sequence with respect to the aforementioned selection matrix. For example,

$$\texttt{aggregate}\left(\begin{bmatrix} 1 & 0 & 0 \\ 0 & 0 & 0 \\ 1 & 1 & 0 \end{bmatrix}, \begin{bmatrix} 10 & 20 & 30 \end{bmatrix}\right) = \begin{bmatrix} 10 & 0 & 15 \end{bmatrix}.$$

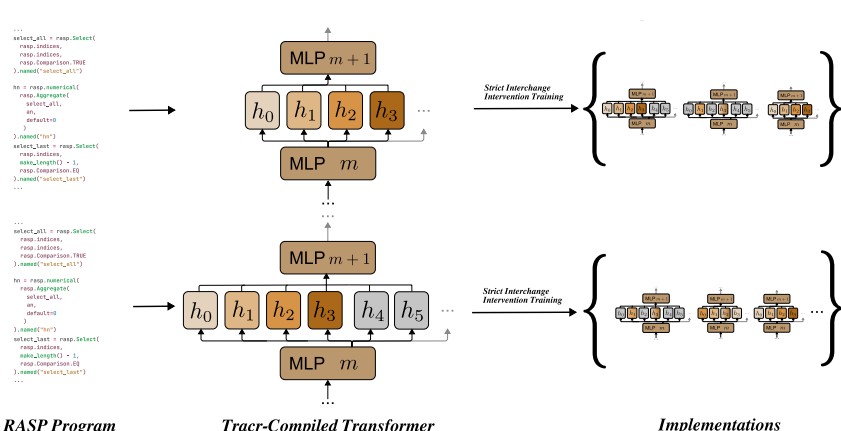

Figure 4: Pipeline for generating implementations for our interpretations (i.e. RASP language models). We first construct RASP programs, then using the procedure introduced in Lindner et al. (2023), we compile these programs into Transformer models. These Transformers exclusively have hard attention. Moreover, their architecture is minimal (containing only the necessary components to fully implement the given RASP program). We then apply the procedure introduced in Gupta et al. (2024) to translate these Tracr-Compiled Transformers into "real" Transformers: ones whose weight distribution matches those trained with stochastic gradient descent; these translated models also contain more

The previous example is directly lifted from Lindner et al. (2023).

**Compiling RASP Programs.** The power of RASP programming lies in its ability to translate any RASP program into a Transformer, a process known as *compilation*. As described in Lindner et al. (2023), this involves a two-stage approach. First, a computational graph is constructed by tracing the *s-ops* in the program, identifying how these operations interact with and modify the residual stream. Elementwise operations are converted into MLP weights, and individual components are heuristically assigned to Transformer layers. For further details, we refer the reader to Lindner et al. (2023).

As observed by Lindner et al. (2023), this compilation through "translation" introduces inefficiencies. Specifically, the heuristic layer-assignment of RASP components results in Transformers that often contain more layers than they need to have. Moreover, since RASP enforces the use of categorical sequences and hard attention (we only allow Boolean predicates) it requires various *s-ops* to lie orthogonal to each other after embedding as Transformer weights. As a result, this leads to a much larger embedding dimension that is usually observed in actual Transformers (Elhage et al., 2022). Thus, Lindner et al. (2023) proposes to compress this dimension through a learned projection matrix. The caveat is that this transformation largely not faithful to the original program (measured through cosine similarity of the outputs at individual layers).

Friedman et al. (2023) takes a different approach, addressing the inherent difficulty of writing RASP programs. To overcome this challenge, the authors propose a method for directly learning RASP programs. This is achieved by constraining the space of learnable weights to those that compile into valid RASP programs, ensuring outputs with categorical variables and hard attention mechanisms. Optimizing over this constrained hypothesis class is performed through a continuous relaxation using the Gumbel distribution (Jang et al., 2017).

**RASP Benchmarks.** Thurnherr and Scheurer (2024) is a dataset of RASP programs that have been generated by GPT-4. It contains 121 RASP programs. Gupta et al. (2024) provides 86 RASP programs and compiled Transformers. The compiled Transformers are claimed to be more realistic than Tracr compiled ones as instead of performing compression using a linear projection, they leverage *strict interchange intervention training* essentially aligning the intervention effects of the compressed and uncompressed model. This is similar in vein to many existing techniques on causal abstraction Otsuka and Saigo (2022); Zennaro (2022); Massidda et al. (2023). In our paper, we leverage the curated dataset Gupta et al. (2024) to craft and compose our interpretations.

## C.2   Strict Interchange Intervention Training

To evaluate our methods, we need a way to verifiably ellicit different mechanisms on the same task. Let us first fix some task. Then, we proceed with the following steps:

1. Using Friedman et al. (2023), we learn several different explicit Transformer programs (source of randomness). We can check that they are different by looking explicitly at the Transformer programs.
2. Using Gupta et al. (2024) and Geiger et al. (2024) to get different mechanistic realizations of this abstract Transformer program.

To generate different mechanistic instantiations of the same interpretation across architectures, we use the following procedure: First, we take a Tracr-Compiled Transformer model and initialize a new random model with at least as many layers and attention heads. Two models share the same interpretation if the Tracr-Compiled transformer is a causal abstraction of our mechanistic instantiation. We leverage this insight by softening Definition D.3 and incorporating it directly into our objective function. For a detailed treatment of this approach, we refer readers to Geiger et al. (2024); Gupta et al. (2024); Sun et al. (2025).

To ensure diversity in our implementations, we vary the architecture hyperparameters significantly: models contain between 2-6 layers, 2-8 attention heads, and embedding dimensions ranging from 32 to 2048. This creates a rich set of instantiations with widely varying model capacities. As a result, many mechanisms in these larger models may not contribute to the core interpretations, leading to substantial interpretive compression.

## C.3   Interpretations for Permutation Detection

To implement this task, we proceed with the following six interpretations. As discussed in Section 3.1, they can be roughly divided into two groups: sort- and counting-based.

**Interpretation 1.** *Sort the sequence*; compute the difference between each element and the next one; (3) Check if all elements of the sequence are equal.

**Interpretation 2.** *Sort the sequence* in descending order; increment each element by its index; check if all elements of the sequence are equal.

**Interpretation 3.** *Sort the sequence*; interleave the list with the same list in reverse order; sum each number with the number next to it; (4) Check if all elements of the list are equal.

**Interpretation 4.** *Sort the sequence*; check if the list contains alternating even and odd elements.

**Interpretation 5.** Check if at least two elements in the list are equal; sum each element with the next one in the list; check if all elements of the list are equal.

**Interpretation 6.** Replace each element with the number of elements less than it in the sequence; check if at least two numbers are the same.

For each of the subroutines of these interpretations, Gupta et al. (2024) implements RASP programs for them. Thus, we simply compose them together to create the resulting interpretations.

## C.4   Generating Implementations for IOI

Herein, we describe how we generate implementations for the IOI task. For each model, we first identify a subset of the model's attention heads that drive functional behavior. We do this through causal mediation analysis (which is well-documented in Wang et al. (2022) and Tigges et al. (2024)). For example, in GPT2-small this corresponds to attention heads (0.1, 0.10, 2.2, 4.11, 5.5, 6.9, etc., formatted as `[layer index].[head index]`). The complement of this set are then components that are "unused" (see Section 2).

For any input like **(a)** we construct a counterfactual input like **(b)** see below:

    (a) `When John went to the store with Mary, John gave a bottle of milk to`
    (b) `When Mary went to the store with John, Mary gave a bottle of milk to`

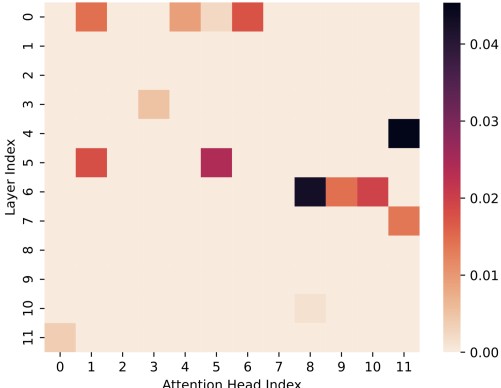

Figure 5: Activation patching results for each attention head. The coloring indicates the % of performance this individual attention head contributes to POS performance.

Then, we generate implementations by ablating the unused components. This is done by running the model on inputs like **(a)** and then for attention head in the ablation subset we patch in the output of the attention head as if it had been run on inputs like **(b)**.

### C.5 FUNCTION VECTORS AND PARTS-OF-SPEECH IDENTIFICATION

Function vectors have been found to drive in-context learning behavior (Todd et al., 2024). We leverage this concept to find the circuit that is responsible for in-context parts-of-speech identification. We use the Penn TreeBank dataset and consider the POS subtask. We sample $n = 1000$ points.

We create counterfactual inputs by shuffling the in-context labels. For example,

Clean Input: `tree:noun, run:verb, quickly:adverb, ire:`
Corrupted Input: `tree:adverb, run:noun, quickly:verb, ire:`

Then, we first run the model on the clean input and store all attention head output activations on the token "ire." We then run the model on the corrupted input and patch in the stored attention head activations from the "ire" token. Finally, we compute the difference in logit difference loss of the generated next tokens before/after this patching operation. The results for each attention head are shown in Figure 5. It seems that attention heads in the early/middle layers are most important for POS (0.1, 0.4, 0.6, 4.11, 5.1, 5.5, 6.8, 6.9, 6.10, 7.11).

Then, we compute the function vector by

$$\frac{1}{|A|} \sum_{a \in A} \sum_{k=1}^{n} a(x_k), \tag{C.1}$$

where $A$ is the set of attention heads that we have deemed important and $x_k$ is our dataset. Essentially, we are averaging the activations across all important attention heads.

## D  FORMAL DEFINITIONS

We provide a glossary (see Table 1 and Table 2 for a summary) for our technical terms and notations in the main text along with their precise, formal definitions.

Table 1: Glossary of terms.

| Term | Intuitive Explanation | Formal Definition |
|---|---|---|
| Causal/Computation Graph | A graph that describes a model's computational dependencies | Definition 4.1 |
| Variables | Nodes in a model's computation graph that store latent computation results | Definition 4.1 |

| | | |
|---|---|---|
| Circuit | A subset of the model's computational graph that on its own can fully recreate the model's input-output behavior | Definition D.1 |
| Mechanistic Explanation | A symbolic algorithm, we can understand, that explains the model's computation | N/A |
| Causal Abstraction | Mapping from low-level variables to higher-level ones that preserve low-level causal relationships | Definition D.3 |
| Interpretation | A causal graph much smaller than the model's computation graph that explains the model's computation graph | Definition D.4 |
| Representations | A causal graph that is a chain that abstracts a model's circuit | Definition 4.3 |
| Alignment | A mapping between two causal graphs that preserves meaningful causal relations between variables | Definition D.3 |
| Intervention | A modification to the model's computation graph | Definition D.2 |
| Alignment Class | A set of alignments that map to the same interpretation | Definition 5.1 |
| Implementation | Sets of circuits that share the same interpretation | Definition 5.1 |
| Interpretive Equivalence | Measure of distance between sets of implementations | Definition 5.2 |
| Interpretive Compression | Measure of diameter of a set of implementations | Definition 5.3 |
| Congruity | Approximation for interpretive equivalence | Definition H.1 |
| Representation Similarity | Extent to which one set of representations can be linearly transformed into the other and vice versa | Definition G.1 |

Table 2: Notations used throughout our formal treatment. This is partially adapted from both Geiger et al. (2025) and Ravfogel et al. (2025).

| Symbol | Meaning |
|---|---|
| $\Sigma$ | An alphabet |
| $\Sigma^\star$ | Set of all finite strings constructed from $\Sigma$ |
| $S$ | A subset of $\Sigma^\star$ |
| $A$ | An interpretation |
| $K$ | A circuit |
| $R$ | A representation |
| $\mathbb{V}$ | Set of variables in a causal model |
| $U$ | Input variable to causal model |
| $\mathbb{F}$ | Functions in causal model that determine variable values |
| $\succ$ | Partial ordering over variables of causal model induced by $\mathbb{F}$ |
| $\mathbb{V}^{\mathbb{F}}(u)$ | Solution of all variables to input $u$ |
| $\boldsymbol{v}_k^{\mathbb{F}}(u)$ | Solution of $\boldsymbol{v}_k$ to input $u$ |
| $\pi$ | An alignment between variables of two causal models |
| $\Pi$ | A class of alignments |
| $\Pi^{-1}(A)$ | The set of implementations of $A$ induced by alignments $\Pi$ |
| $d_{\text{interp}}$ | Interpretive equivalence |
| $d_{\text{H}}$ | Hausdorff distance |
| $\kappa(A, K, \Pi)$ | Interpretive compression |
| $d_{\text{repr}}$ | Representation similarity |
| $\text{Lip}(f)$ | Lipschitz constant of $f$ |
| $\overset{\text{i.i.d.}}{\sim}$ | Sample i.i.d from some distribution |

**Definition D.1** (Circuit). An $m$-circuit of $h$ on $S$ is a causal graph with $m$ components that satisfies four properties:

1. Input variable is $S$-valued: $U \in S$
2. Hidden variables are real-valued: for each $\boldsymbol{v}_k \in \mathbb{V}$, $\boldsymbol{v} \in \mathbb{R}^{n_k}$
3. Existence of a terminal output variable: $\boldsymbol{v}_{\text{out}} \in \mathbb{V}$ such that $\boldsymbol{v}_{\text{out}} \in \mathbb{R}^d$, and there does not exist $\boldsymbol{v}' \in \mathbb{V}$ where $\boldsymbol{v}_{\text{out}} \succ \boldsymbol{v}'$
4. Sufficient description of $h$ on $S$: for each $s \in S$, $\boldsymbol{v}_{\text{out}}^{\mathbb{F}}(s) = h(s)$

*Remark.* For any $h$ and $m \geq 1$, an $m$-circuit of $h$ must exist by the trivial causal model. To see this, let $U \in S$ and $v_{\text{out}} \triangleq h(U)$. Then, for any $1 \leq i \leq m-1$, define $v_i \triangleq U \mapsto 1$. In other words, we construct a causal graph with effectively one variable, letting the other hidden variables be constant functions.

*Remark.* $m$-circuits are also not unique. For instance, a joint change of bases to any $v_k \in \mathbb{V} \setminus v_{\text{out}}$ and $f_k$ results in a new causal model that preserves functional faithfulness to $h$ (Park et al., 2024; Geiger et al., 2024).

**Definition D.2** (Intervention). Let $(\mathbb{V}, U, \mathbb{F}, \succ)$ be a causal model. An **intervention** of $v_k \in \mathbb{V}$, $\text{do}(v_k \leftarrow \tilde{f})$, redefines $v_k \triangleq \tilde{f}(\tilde{Pa}(v_k), U)$, for $\tilde{Pa}(v_k) \subset \{v \in \mathbb{V} : v \succ v_k\}$. We denote the set of interventions on $v_k$ as $\mathscr{I}(v_k)$.

In other words, we replace the process used to derive $v_k$'s value, $f$, with a new function $\tilde{f}$. Given any circuit, one type of concrete intervention is **activation patching**. For a set of hidden variables, we patch its solutions using values derived from some other input (Vig et al., 2020; Wang et al., 2022; Heimersheim and Nanda, 2024, *inter alia*). Interventions often serve as a check of consistency between circuits and their interpretations (Beckers and Halpern, 2019; Geiger et al., 2021). We define this notion next.

**Definition D.3** (Abstraction). Let $K_1 \triangleq (\mathbb{V}, U, \mathbb{F}, \succ_1), K_2 \triangleq (\tilde{\mathbb{V}}, \tilde{U}, \tilde{\mathbb{F}}, \succ_2)$ be causal models. $K_2$ **abstracts** $K_1$ if there exists surjective mappings $\pi : \text{SubsetsOf}(\mathbb{V}) \rightarrow \tilde{\mathbb{V}}, \omega : \mathscr{I}(\mathbb{V}) \rightarrow \mathscr{I}(\tilde{\mathbb{V}})$ that satisfies for all $v_k \in \mathbb{V}, \tilde{v}_k \in \tilde{\mathbb{V}}, \text{do}(v_k \leftarrow f) \in \mathscr{I}(v_k)$:

$$\tilde{v}_k = \pi \left( \bigcup_{v_k \in \pi^{-1}(\tilde{v}_k)} f_k(Pa(v_k), U) \right), \tag{D.1}$$

$$\pi(\text{do}(v_k \leftarrow f)) = \text{do}(\pi(v_k) \leftarrow \omega(f)). \tag{D.2}$$

We call $\pi$ an **alignment**.

Equation (D.1) describes an observational consistency constraint: if $K_2$ abstracts $K_1$, then by only observing the hidden variables of $K_1$ we can infer all hidden values of $K_2$. Viewed under this lens, the alignment $\pi$ projects the fine-grained variables $\mathbb{V}$ to the coarse-grained ones $\tilde{\mathbb{V}}$. On the other hand, Equation (D.2) is an intervention consistency constraint—stating that $\pi$ and $\omega$ must commute. In words, this means that if we apply the intervention $\text{do}(v_k \leftarrow f)$ to yield a new causal model and then abstract that causal model through alignment $\pi$, this is equivalent to abstracting the causal model and then applying the transformed intervention on the high-level model. This implies that all of the causal relationships between variables in $K_2$ can be constructed by relationships between variables in $K_1$. Under this light, an interpretation of a circuit is simply an abstraction.

**Definition D.4** (Interpretation). A causal model $A \triangleq (\mathbb{V}^{\star}, U, \mathbb{F}^{\star}, \succ)$ that abstracts a circuit $K \triangleq (\mathbb{V}, U, \mathbb{F}, \succ)$ is an $\eta$-faithful interpretation of $K$ if there exists an output variable $v_{\text{out}}^{\star} \in \mathbb{V}^{\star}$ such that $\|v_{\text{out}}^{\star, \mathbb{F}} - v_{\text{out}}^{\mathbb{F}}\| < \eta$ for the output variable $v_{\text{out}} \in \mathbb{V}$.

*Remark.* Interpretations are abstractions of *circuits*. However, Definition D.4 does not rely on the fact that circuits have real-valued hidden variables. Since interpretations also have an output variable, one could further construct an interpretation of an interpretation (Proposition F.2).

*Remark.* For any circuit, $K$ is an interpretation of itself (under the identity abstraction), albeit this is not very useful.

*Remark.* We *do not* require that $A$'s variables be real-valued, nor do we require that there exists a metric over the solutions of $A$ (in contrast to $K$). This is compatible with the conventions in mechanistic interpretability where often an interpretation is a symbolic description of $K$ that may not necessarily be performing numeric computation.

# E    INTERPRETATIONS MUST BE ABSTRACTION DEPENDENT

Our notion of an interpretation is dependent on a fixed level of abstraction (the variable set $\mathbb{V}$). This requirement may seem stringent, especially since we colloquially think of interpretations (or algorithms) as abstraction-independent entities. Further, abstraction-*dependent* interpretations immediately give rise to *non-identifiability*, where a single interpretation may correspond many

implementations (at different levels of abstraction) *and* a single circuit may also correspond many interpretations, each at a different level of abstraction.

In this section, we argue that non-identifiability is a necessary price we must pay for a meaningful comparison between interpretations. Our main result here is that if defined over an existence statement of $\mathbb{V}$ interpretive equivalence must collapse to functional equivalence.

**Theorem E.1.** *For $i \in \{1,2\}$, let $S \subset \Sigma^\star$ and $h_i$ be a language model such that $h_1(S) = h_2(S)$. Suppose that $h_i$ admit a circuit $K_i \triangleq (\mathbb{V}_i, U, \mathbb{F}_i, \succ)$. Then,*

1. *For $i \in \{1,2\}$, there exists an abstraction of $K_i$, $A_i \triangleq (\mathbb{V}_A, U, \mathbb{F}_i^A, \succ)$ such that there is a* **bijective translation** [12] *(Definition E.1) between $A_1, A_2$.*
2. *There is a language model $h^\star$ with circuit $K^\star$ such that $h^\star(S) = h_1(S) = h_2(S)$ and $K_1, K_2$ are abstractions of $K^\star$.*
3. *If both $K_1, K_2$ are not minimal (i.e. there exists at least one causal variable such that any intervention does not affect its output), then there exists a bijective translation between $K_1, K_2$.*

Since we do not require circuits to be minimal (see Section 4), statement **(3)** implies that we must restrict the set of permissible alignments. Otherwise, any non-minimal circuit is 0-interpretively equivalent to any other non-minimal circuit (see Proposition F.4).

Recall that for any causal model $K \triangleq (\mathbb{V}, U, \mathbb{F}, \succ)$, given an input $U \leftarrow u$, there exists a solution to each of its variables which we denote as $\mathbb{V}^{\mathbb{F}}(u)$. If $K$ is an $m$-circuit, then

$$\mathbb{V}^{\mathbb{F}}(u) \in \Sigma^\star \times \mathbb{R}^{n_1} \times \ldots \times \mathbb{R}^{n_m} \times \mathbb{R}^{|\Sigma|}. \tag{E.1}$$

For brevity, we slightly abuse the notation and denote $\mathbb{V}^{\mathbb{F}} \triangleq \mathbb{R}^{n_1} \times \ldots \times \mathbb{R}^{n_m} \times \mathbb{R}^{|\Sigma|}$. In words, a **bijective translation** is a causally-consistent, bijective mapping between the solutions of two causal models. Essentially, if one model is a bijective translation of another, we can think of it simply as a relabeling of the other one.

**Definition E.1.** *For $i \in \{1,2\}$, let $K_i \triangleq (\mathbb{V}_i, U, \mathbb{F}_i, \succ)$ be a causal model. For any $u \in S$, $\tau : \mathbb{V}_1^{\mathbb{F}_1} \rightarrow \mathbb{V}_2^{\mathbb{F}_2}$ is a bijective translation if $\tau$ satisfies Equation (D.1) and there exists $\omega : \mathscr{I}(\mathbb{V}_1) \rightarrow \mathscr{I}(\mathbb{V}_2)$ that satisfies Equation (D.2).*

**Proposition E.2.** *For any $n, m \in \mathbb{Z}^+$, there exists a bijection between $\mathbb{R}^n$ and $\mathbb{R}^m$.*

We are now ready for the proof of Theorem E.1.

*Proof.* We prove all claims constructively.

1. For $i \in \{1,2\}$, let $A_i$ be the trivial causal model associated with $h_i$: it has two causal variables $U, v_{\text{out}}$ such that $v_{\text{out}} \triangleq h_i(U)$. Consider an an alignment map $\pi_i$ such that $\pi_i(v_{\text{out}}) = v_{\text{out}}$, $\pi_i(U) = U$, and for all other $v \in \mathbb{V}_i$, $\pi_i(v) = \perp$ (i.e. $\pi$ is undefined on other variables in $\mathbb{V}_i$). Define the intervention map $\omega$ to be the identity if and only if $\text{do}(v_k \leftarrow f) \in \mathscr{I}(\mathbb{V}_i)$ is an intervention on $U$ or $v_{\text{out}}$. Then, it follows that the only valid interventions defined by $\omega$ are those directly on the input and output variables. It is easy to see that both Equation (D.1) and (D.2) hold. Since $h_1 = h_2$, $A_1, A_2$ constructed in this way are the same causal model. Thus, the identity map forms a trivial bijective translation between them.

2. Let $K^\star \triangleq (\mathbb{V}, U, \mathbb{F}, \succ)$ such that $\mathbb{V} = \mathbb{V}_1 \cup \mathbb{V}_2 \cup \{U, v_{\text{out}}\}$. Construct $\mathbb{F}$ such that it runs $K_1, K_2$ in parallel and then averages their output. Concretely, let $v_{\text{out}}^1, v_{\text{out}}^2 \in \mathbb{V}$ be the output variables of $\mathbb{V}_1, \mathbb{V}_2$, respectively. Also let $U^1, U^2 \in \mathbb{V}$ be their input variables. Define

$$Pa(U_1) = Pa(U_2) = \{U\} \qquad Pa(v_{\text{out}}) = \{v_{\text{out}}^1, v_{\text{out}}^2\},$$

and their transition functions

$$f_{U_1}(U) = f_{U_2}(U) = U \qquad f_{v_{\text{out}}}(\{v_{\text{out}}^1, v_{\text{out}}^2\}, U) = \frac{1}{2}\left(f_{v_{\text{out}}^1}(Pa(v_{\text{out}}^1), U) + f_{v_{\text{out}}^2}(Pa(v_{\text{out}}^2), U)\right).$$

Clearly, $K^\star$ is a valid circuit for some language model $h = h_1 = h_2$. Let $\pi_1, \pi_2$ be the identity alignments from $K^\star \rightarrow K_i$. By the same constructive above for $\omega$, $K_1, K_2$ must be both be an abstraction for $K^\star$.

---

[12]In general, a bijective translation is neither stronger nor weaker than an abstraction (Geiger et al., 2025).

3. Let $\tau$ be the identity function that maps $\boldsymbol{v}_{\text{out}}^1 \in \mathbb{V}_1 \to \boldsymbol{v}_{\text{out}}^2 \in \mathbb{V}_2$. By Definition 4.2, for $i \in \{1, 2\}$, any variable $\boldsymbol{v}_k \in \mathbb{V}_i \setminus \{U, \boldsymbol{v}_{\text{out}}^i\}$ takes on values in $\mathbb{R}^{m_k}$ for $m_k \in \mathbb{Z}^+$. By assumption, for both models $K_1, K_2$, there exists at least one causal variable that is "unused." Denote this variable as $\boldsymbol{v}_{\text{unused}}^i$. By Proposition E.2, there exists a bijection that maps $\sigma : \mathbb{V}_1 \setminus \{U, \boldsymbol{v}_{\text{out}}, \boldsymbol{v}_{\text{unused}}^1\} \to \boldsymbol{v}_{\text{unused}}^2$. Composing $\sigma \circ \tau$ yields a bijection from $\mathbb{V}_1^{\mathbb{F}_1} \to \mathbb{V}_2^{\mathbb{F}_2}$. It is easy to see that this is a bijective translation which satisfies Equation (D.1) and (D.2). By symmetry, the other direction also holds.

$\square$

# F  Properties of Interpretive Equivalence and Compression

As discussed in Section 5, an interpretation of a model need not be a lossless description of that model. For many applications an interpretable explanation is necessarily lossy with respect to the learned model. In this sense, an abstraction alone is too restrictive as it requires any interpretation to preserve the exact functional behavior of the model. Herein, we explore some properties of interpretive equivalence, compression and illustrate their similarities and differences with causal abstraction.

We first show that both implementations and interpretations compose. That is, *an implementation of an implementation is also an implementation* and *an interpretation of an interpretation is also an interpretation*.

**Lemma F.1.** *For $i \in \{1, 2, 3\}$, let $K_i \triangleq (\mathbb{V}_i, U, \mathbb{F}_i, \succ_i)$ be causal models. Let $\pi_{1,2}$ be an alignment from $K_1 \to K_2$ and $\pi_{2,3}$ an alignment from $K_2 \to K_3$. Then, $K_3$ is an abstraction of $K_1$ with an alignment map $\pi_{2,3} \circ \pi_{1,2}$.*

*Remark* F.1 (On Composing Alignments). We are liberally abusing the notation here. Technically, the composition $\pi_{2,3} \circ \pi_{1,2}$ is not well defined since the alignments have incompatible types: $\pi_{1,2} :$ SubsetsOf$(\mathbb{V}_1) \to \mathbb{V}_2$ and $\pi_{2,3} :$ SubsetsOf$(\mathbb{V}_2) \to \mathbb{V}_3$. We write $\pi_{2,3} \circ \pi_{1,2}$ to mean the process of starting from the causal graph $K_1$ applying the alignment $\pi_{1,2}$ to yield a new causal graph $K_2$ and then applying the alignment $\pi_{2,3}$ to yield the causal graph $K_3$. This is possible because an alignment alone is sufficient to induce an abstraction (see Geiger et al. 2025, Remark 32 for a construction).

*Proof.* We first check Equation (D.1). Since $\pi_{2,3}$ is an alignment that enables $K_2$ to abstract $K_1$, for any $\boldsymbol{v}_k^3 \in \mathbb{V}_3$,

$$\boldsymbol{v}_k^3 = \pi_{2,3}\left(\bigcup_{\boldsymbol{v}_k^2 \in \pi_{2,3}^{-1}(\boldsymbol{v}_k^3)} f_k^2(Pa(\boldsymbol{v}_k^2), U)\right).$$

By Definition 4.1, $\boldsymbol{v}_k^2 = f_k^2(Pa(\boldsymbol{v}_k^2), U)$. Then, because $K_2$ is an abstraction of $K_1$ through the alignment map $\pi_{1,2}$, it follows that

$$\boldsymbol{v}_k^3 = \pi_{2,3}\left(\bigcup_{\boldsymbol{v}_k^2 \in \pi_{2,3}^{-1}(\boldsymbol{v}_k^3)} \pi_{1,2}\left(\bigcup_{\boldsymbol{v}_k^1 \in \pi_{1,2}^{-1}(\boldsymbol{v}_k^2)} f_k^1(Pa(\boldsymbol{v}_k^1), U)\right)\right),$$

$$= (\pi_{2,3} \circ \pi_{1,2})\left(\bigcup_{\boldsymbol{v}_k^2 \in \pi_{2,3}^{-1}(\boldsymbol{v}_k^3)}\left(\bigcup_{\boldsymbol{v}_k^1 \in \pi_{1,2}^{-1}(\boldsymbol{v}_k^2)} f_k^1(Pa(\boldsymbol{v}_k^1), U)\right)\right),$$

$$= (\pi_{2,3} \circ \pi_{1,2})\left(\bigcup_{\boldsymbol{v}_k^1 \in (\pi_{2,3} \circ \pi_{1,2})^{-1}(\boldsymbol{v}_k^3)} f_k^1(Pa(\boldsymbol{v}_k^1), U)\right).$$

It remains to show that the mappings between interventions holds. Denote $\omega_{1,2} : \mathscr{I}(\mathbb{V}_1) \to \mathscr{I}(\mathbb{V}_2)$ and $\omega_{2,3} : \mathscr{I}(\mathbb{V}_2) \to \mathscr{I}(\mathbb{V}_3)$. We claim that Equation (D.2) is satisfied with $\omega_{2,3} \circ \omega_{1,2}$. Since $\omega_{1,2}$ and $\omega_{2,3}$ satisfy Equation (D.2),

$$\pi_{2,3}(\text{do}(\boldsymbol{v}_k^2 \leftarrow f)) = \text{do}(\pi_{2,3}(\boldsymbol{v}_k^2) \leftarrow \omega_{2,3}(f)),$$

$$\pi_{2,3}(\pi_{1,2}(\mathrm{do}(v_k^1 \leftarrow f))) = \mathrm{do}(\pi_{2,3}(\pi_{1,2}(v_k^1)) \leftarrow \omega_{2,3}(\omega_{1,2}(f))),$$
$$(\pi_{2,3} \circ \pi_{1,2})(\mathrm{do}(v_k^1 \leftarrow f)) = \mathrm{do}((\pi_{2,3} \circ \pi_{1,2})(v_k^1) \leftarrow (\omega_{2,3} \circ \omega_{1,2})(f)),$$

where $v_k^2 \in \mathbb{V}_2$, $v_k^1 \in \pi_{1,2}^{-1}(v_k^2)$, and $f$ are chosen arbitrarily. $\qquad\square$

**Proposition F.2.** *Let $K \triangleq (\mathbb{V}, U, \mathbb{F}, \succ)$ be a circuit. Let $A \triangleq (\mathbb{V}_A, U, \mathbb{F}_A, \succ)$ be an $\eta$-faithful interpretation of $K$ through an alignment $\pi$. Let $A^\star \triangleq (\mathbb{V}_A^\star, U, \mathbb{F}_A^\star, \succ)$ be an $\eta$-faithful interpretation of $A$ through an alignment $\pi^\star$. Then, $A^\star$ is an $2\eta$-faithful interpretation of $K$ through an alignment $\pi^\star \circ \pi$.*

*Proof.* There exists an alignment $\pi$ from $K \to A$ and an alignment $\pi^\star A \to A^\star$. By Lemma F.1, $A^\star$ is an abstraction of $K$ with the alignment map $\pi^\star \circ \pi$. There exists a variable $v_{\mathrm{out}}^{A^\star} \in \mathbb{V}_A^\star$, $v_{\mathrm{out}}^A \in \mathbb{V}_A$ such that $\left\| v_{\mathrm{out}}^{A^\star} - v_{\mathrm{out}}^A \right\| < \eta$ and a variable $v_{\mathrm{out}} \in \mathbb{V}$ such that $\left\| v_{\mathrm{out}}^A - v_{\mathrm{out}} \right\| < \eta$. By triangle inequality, $\left\| v_{\mathrm{out}}^{A^\star} - v_{\mathrm{out}} \right\| < 2\eta$. $\qquad\square$

**Proposition F.3.** *Let $K \triangleq (\mathbb{V}, U, \mathbb{F}, \succ)$ be a circuit. Let $A$ be an $\eta$-interpretation of $K$ and $\Pi$ be an alignment class that maps $K$'s variables onto $A$'s. Let $A^\star$ be an $\eta$-interpretation of $A$ and $\Pi_\star$ be an alignment class that maps $A$'s variables onto $A^\star$'s. Then, $\Pi^{-1}(\Pi_\star^{-1}(A_\star))$ are implementations of $A^\star$.*

*Proof.* The proof is trivial and directly follows from Proposition F.2. $\qquad\square$

We now relate interpretive equivalence to causal abstraction and show necessary and sufficient conditions for interpretive equivalence using abstractions.

For $i \in \{1, 2\}$, let $K \triangleq (\mathbb{V}, U, \mathbb{F}, \succ)$ be circuits with $\eta$-interpretations $A_i \triangleq (\mathbb{V}_i^A, U, \mathbb{F}_i^A, \succ)$. Let $\Pi_i$ be alignment classes that map $K_i$'s variables onto $A_i$'s variables.

**Proposition F.4.** *Suppose $A_1$ is an abstraction of $A_2$ and vice versa through alignment maps $\pi_{1,2}, \pi_{2,1}$. $A_1, A_2$ are 0-interpretively equivalent if $\pi_{2,1} \circ \Pi_1 \subset \Pi_2$ and $\pi_{1,2} \circ \Pi_2 \subset \Pi_1$.*

*Proof.* We show that $\Pi_1^{-1}(A_1) \subset \Pi_2^{-1}(A_2)$. Take any $F \in \Pi_1^{-1}(A_1)$. By Definition 5.1, there exists $\pi_1 \in \Pi_1$ that forms an abstraction from $(\mathbb{V}, U, F, \succ) \to (\mathbb{V}_1^A, U, \mathbb{F}_1^A, \succ)$. By Lemma F.1, $\pi_{2,1} \circ \pi_1$ is an alignment that forms an abstraction from $(\mathbb{V}, U, F, \succ) \to (\mathbb{V}_2^A, U, \mathbb{F}_2^A, \succ)$. Since $\pi_{2,1} \circ \pi_1 \in \Pi_2^{-1}(A_2)$, this implies that $F \in \Pi_2^{-1}(A_2)$. Applying the same argument in the other direction, $\Pi_2^{-1}(A_2) \subset \Pi_1^{-1}(A_1)$. Therefore, $\Pi_1^{-1}(A_1) = \Pi_2^{-1}(A_2)$ and by Definition 5.2, $A_1, A_2$ are 0-interpretively equivalent. $\qquad\square$

**Proposition F.5.** *Let $\Pi_i^{-1}(A_i)$ be closed. Suppose $A_1, A_2$ are 0-interpretively equivalent. Suppose there exists $\pi_i \in \Pi_i$ that each admits a **section**, $s_i : \mathbb{V}_i^A \mapsto \mathrm{SubsetsOf}(\mathbb{V}_i)$, such that $\pi_i(s_i(v)) = v$, for any $v \in \mathbb{V}_i^A$. Suppose that there exists some $\pi_1', \pi_2'$ such that $\pi_2' \circ s_1 \circ \pi_1 \in \Pi_2$ and $\pi_1' \circ s_2 \circ \pi_2 \in \Pi_1$. Then, $A_1$ is an abstraction of $A_2$ and vice versa with alignment maps $\pi_2' \circ s_1, \pi_1' \circ s_2$.*

*Proof.* Since $\Pi_i^{-1}(A_i)$ is closed, Definition 5.2 implies that $\Pi_1^{-1}(A_1) = \Pi_2^{-1}(A_2)$. For brevity, we denote $\pi_{2,1} \triangleq \pi_2' \circ s_1$. Note that $\pi_{2,1} : \mathbb{V}_1^A \to \mathbb{V}_2^A$, we can easily extend this to subsets $V \subset \mathbb{V}_1^A$ by

$$\pi_{2,1}(V) \triangleq \pi_2' \left( \bigcup_{v \in V} s_1(v) \right). \tag{F.1}$$

Now, we refer to $\pi_{2,1}$ as Equation (F.1). We first show that $\pi_{2,1}$ satisfies Equation (D.1). Fix $v_2 \in \mathbb{V}_2^A$. By Definition 4.1,

$$v_2 \triangleq f_{v_2}(Pa(v_2), U). \tag{F.2}$$

By assumption, $\pi_{2,1} \circ \pi_1 \in \Pi_2$. Thus, $\pi_{2,1} \circ \pi_1$ is an alignment map that enables an abstraction from $K \to A_2$, we can rewrite Equation (F.2) as

$$v_2 = (\pi_{2,1} \circ \pi_1) \left( \bigcup_{v_k \in (\pi_{2,1} \circ \pi_1)^{-1}(v_2)} f_{v_k}(Pa(v_k), U) \right), \tag{F.3}$$

$$= (\pi_{2,1} \circ \pi_1) \left( \bigcup_{\boldsymbol{v}_k \in \pi_{2,1}^{-1}(\boldsymbol{v}_2)} \bigcup_{\boldsymbol{v} \in \pi_1^{-1}(\boldsymbol{v}_k)} f_{\boldsymbol{v}}(Pa(\boldsymbol{v}), U) \right), \tag{F.4}$$

$$= \pi_{2,1} \left( \pi_1 \left( \bigcup_{\boldsymbol{v}_k \in \pi_{2,1}^{-1}(\boldsymbol{v}_2)} \bigcup_{\boldsymbol{v} \in \pi_1^{-1}(\boldsymbol{v}_k)} f_{\boldsymbol{v}}(Pa(\boldsymbol{v}), U) \right) \right), \tag{F.5}$$

$$= \pi_{2,1} \left( \bigcup_{\boldsymbol{v}_k \in \pi_{2,1}^{-1}(\boldsymbol{v}_2)} \pi_1 \left( \bigcup_{\boldsymbol{v} \in \pi_1^{-1}(\boldsymbol{v}_k)} f_{\boldsymbol{v}}(Pa(\boldsymbol{v}), U) \right) \right), \tag{F.6}$$

$$= \pi_{2,1} \left( \bigcup_{\boldsymbol{v}_k \in \pi_{2,1}^{-1}(\boldsymbol{v}_2)} f_{\boldsymbol{v}_k}(Pa(\boldsymbol{v}_k), U) \right). \tag{F.7}$$

The last equality follows from the fact that $\pi_1$ is an alignment. Therefore, $\pi_{2,1}$ satisfies Equation (D.1). By the same argument, this also holds for $\pi_{1,2}$. It remains to check Equation (D.2). Since $\pi_2', \pi_1'$ are alignments, they admit $\omega_2', \omega_1'$ that satisfy Equation (D.2). Let $t_2, t_1$ be sections of $\omega_2', \omega_1'$, respectively. We claim that $\omega_{2,1} \triangleq \omega_2' \circ t_1$ satisfies Equation (D.2). Since $\pi_{2,1} \circ \pi_1$ is an alignment, and take $\boldsymbol{v}_k \in \mathbb{V}$,

$$(\pi_{2,1} \circ \pi_1)(\mathrm{do}(\boldsymbol{v}_k \leftarrow f)) = \mathrm{do}((\pi_{2,1} \circ \pi_1)(\boldsymbol{v}_k) \leftarrow (\omega_{2,1} \circ \omega_1')(f)),$$
$$\pi_{2,1}(\pi_1(\mathrm{do}(\boldsymbol{v}_k \leftarrow f))) = \mathrm{do}(\pi_{2,1}(\pi_1(\boldsymbol{v}_k)) \leftarrow \omega_{2,1}(\omega_1'(f))),$$
$$\pi_{2,1}(\mathrm{do}(\pi_1(\boldsymbol{v}_k) \leftarrow \omega_1'(f))) = \mathrm{do}(\pi_{2,1}(\pi_1(\boldsymbol{v}_k)) \leftarrow \omega_{2,1}(\omega_1'(f))).$$

$\mathrm{do}(\pi_1(\boldsymbol{v}_k) \leftarrow \omega_1'(f)) \in \mathscr{I}(\mathbb{V}_1^A)$ and $\pi_1(\boldsymbol{v}_k) \in \mathbb{V}_1^A$. Thus, through relabeling, $\boldsymbol{v}_k^1 \triangleq \pi_1(\boldsymbol{v}_k)$ and $f_1 \triangleq \omega_1'(f)$,

$$\pi_{2,1}(\mathrm{do}(\boldsymbol{v}_k^1 \leftarrow f_1)) = \mathrm{do}(\pi_{2,1}(\boldsymbol{v}_k^1) \leftarrow \omega_{2,1}(f_1)).$$

And, Equation (D.2) holds. $\qquad \square$

## G  PROOFS FOR REPRESENTATION SIMILARITY AND INTERPRETIVE EQUIVALENCE

Suppose we have two circuits over a shared task $S$ for models $h_1, h_2$ defined over different neural architectures. These circuits we notate as $K_1 \triangleq (\mathbb{V}_1, U, \mathbb{F}_1, \succ)$ and $K_2 \triangleq (\mathbb{V}_2, U, \mathbb{F}_2, \succ)$, respectively. Let us also assume each circuit admits an $(L_i, \delta_i)$-representation $R_i \triangleq (\mathbb{H}_i, U, \mathbb{F}_i, \succ)$ through alignment maps $\rho_i$. First, we define a notion of distance between these two representations. We note that since the representations themselves are a function of the input (recall Definition 4.1), we are essentially defining a distance in a function space.

**Definition G.1** (Representation Similarity). Denote by $H_i \triangleq \mathrm{concat}(\mathbb{H}_i^{\mathbb{F}_i})$ to be the direct sum (concatenation) of all hidden variables in $\mathbb{H}_i$. Then, representation similarity[13] between $R_1$ and $R_2$ is

$$d_{\mathrm{repr}}(R_1, R_2) \triangleq \max \left( \inf_{\|A\|_{\mathrm{op}} \leq 1} \|AH_1 - H_2\|, \inf_{\|B\|_{\mathrm{op}} \leq 1} \|H_1 - BH_2\| \right), \tag{G.1}$$

where $A, B$ are linear operators and $\|\cdot\|$ is a direct-sum norm.

We first show that representation similarity (Definition G.1) is a pseudometric.

**Lemma G.1.** *For any representations $R_1, R_2, R_3$, $d_{\mathrm{repr}}$ satisfies*

1. $d_{\mathrm{repr}}(R_1, R_1) = 0$
2. $d_{\mathrm{repr}}(R_1, R_2) = d_{\mathrm{repr}}(R_2, R_2)$
3. $d_{\mathrm{repr}}(R_1, R_3) \leq d_{\mathrm{repr}}(R_1, R_2) + d_{\mathrm{repr}}(R_2, R_3)$

---

[13]Our formulation is inspired by Chan et al.'s (2024) notion of representation alignment between language encoders. Although Definition G.1 is not a true similarity, this naming stays consistent with the literature (Kornblith et al., 2019).

*Proof.* The first two properties are trivial. So, it remains to show triangle inequality. Let $H_i$ be the concatenation of hidden variables in $R_i$. Let $\varphi_{1,2}, \varphi_{2,3}$ be arbitrary operators where $\|\varphi_{1,2}\|_{\mathrm{op}}, \|\varphi_{2,3}\|_{\mathrm{op}} \leq 1$ that map $H_2 \to H_1$ and $H_3 \to H_2$, respectively. Then,

$$
\begin{aligned}
\|H_1 - \varphi_{1,2} \circ \varphi_{2,3} H_3\| &= \|H_1 - \varphi_{1,2} H_2 + \varphi_{1,2} H_2 - \varphi_{1,2} \circ \varphi_{2,3} H_3\|, \\
&\leq \|H_1 - \varphi_{1,2} H_2\| + \|\varphi_{1,2} H_2 - \varphi_{1,2} \circ \varphi_{2,3} H_3\|, \\
&\leq \|H_1 - \varphi_{1,2} H_2\| + \|\varphi_{1,2}\|_{\mathrm{op}} \|H_2 - \varphi_{2,3} H_3\| \\
&\leq \|H_1 - \varphi_{1,2} H_2\| + \|H_2 - \varphi_{2,3} H_3\|.
\end{aligned}
$$

Taking infimum over $\varphi_{1,2}, \varphi_{2,3}$ gives the desired result. $\qquad\square$

**Definition G.2.** Let $(Z,d)$ be a (pseudo)metric space. Let $S \subset Z$. The **diameter** of $S$, denoted diameter$(S)$ is defined as

$$
\mathrm{diameter}(S) := \sup_{a,b \in S} d(a,b). \tag{G.2}
$$

**Lemma G.2.** *Let $(Z,d)$ be a (pseudo)metric space and $d_{\mathrm{H}}$ be its Hausdorff distance. Then, for $A, B \subset Z$ and $a \in A, b \in B$,*

$$
d_{\mathrm{H}}(A,B) \leq \mathrm{diameter}(A) + \mathrm{diameter}(B) + d(a,b).
$$

*Proof.* Fix any $a' \in A$. Then,

$$
\begin{aligned}
\inf_{b' \in B} d(a',b') &\leq d(a',b), \\
&\leq d(a',a) + d(a,b), \\
&\leq \mathrm{diameter}(A) + d(a,b).
\end{aligned}
$$

By symmetry, for any $b' \in A$, it follows that $\inf_{a' \in A} d(a',b') \leq \mathrm{diameter}(B) + d(a,b)$. Therefore, by the definition of the Hausdorff distance, we yield the desired bound. $\qquad\square$

**Definition G.3.** Let $K_1 \triangleq (\mathbb{V}, U, F_1, \cdot), K_2 \triangleq (\mathbb{V}, U, F_2, \cdot)$ be two arbitrary circuits defined over variables $\mathbb{V}$. Suppose some pseudometric $d : \mathbb{V} \times \mathbb{V} \to \mathbb{R}^{\geq 0}$. Let $R \triangleq (\mathbb{H}, U, \cdot, \prec)$ be representation spaces and $\rho : \mathrm{SubsetsOf}(\mathbb{V}) \to \mathbb{H}$ an alignment map. Denote $R_i$ the representations of $K_i$ constructed through $\rho$. We say that $\rho$ is Lipschitz continuous with Lipschitz constant $\mathrm{Lip}(\rho)$ if

$$
d_{\mathrm{repr}}(R_1, R_2) \leq \mathrm{Lip}(\rho) d(F_1, F_2). \tag{G.3}
$$

**Lemma G.3.** *For $i \in \{1,2\}$ let $R_i$ be $(\cdot, \delta_i)$-representations for models $h_i$. Suppose that $\|h_1 - h_2\| < \Delta$. Then, $d_{\mathrm{repr}}(R_1, R_2) \leq \delta_1 + \delta_2 + \Delta$.*

*Proof.* Let $H_i$ be the concatenation of all hidden variables in $R_i$. Since $R_i$ is a $(L_i, \delta_i)$-representation, there must exist maps $A_i, B_i$ such that

$$
\|A_i H_i - h_i\| < \delta_i \qquad \text{and} \qquad \|H_i - B_i h_i\| < \delta_i.
$$

To see this, we know that for each hidden layer $k$, there exists $A_{i,k}$ such that $\|A_{i,k} H_{i,k} - h_i\| < \delta_i$. Then,

$$
L_i \delta_i \geq \sum_{k=1}^{L_i} \|A_{i,k} H_{i,k} - h_i\| \geq \left\| \sum_{k=1}^{L_i} (A_{i,k} H_{i,k} - h_i) \right\| = \left\| \sum_{k=1}^{L_i} A_{i,k} H_{i,k} - L_i h_i \right\|. \tag{G.4}
$$

Define a single (block diagonal) linear map $A_i$ that acts on $H_i = (H_{i,1}, \ldots, H_{i,L_i})$ where

$$
\|A_i H_i\| \triangleq \frac{1}{L_i} \sum_{k=1}^{L_i} \|A_{i,k} H_{i,k}\|.
$$

With the $\ell_\infty$ direct-sum norm on $\bigoplus_k \mathbb{R}^{n_k}$ and $\|A_i\|_{\mathrm{op}} \leq 1$. Substituting $A_i$ into Equation (G.4), we have that $\|A_i H_i - h_i\| < \delta_i$. Repeating same construction for $B_i$ yields the "decoder" inequality.

Consider $\inf_{A_0} \|A_0 H_1 - H_2\|$,

$$
\inf_{A_0} \|A_0 H_1 - H_2\| \leq \|B_2 A_1 H_1 - H_2\|,
$$

$$\leq \|B_2 A_1 H_1 - B_2 h_2\| + \|B_2 h_2 - H_2\|,$$
$$\leq \|B_2\|_{\text{op}} (\|A_1 H_1 - h_1\| + \|h_1 - h_2\|) + \delta_2,$$
$$\leq \delta_1 + \delta_2 + \Delta.$$

The same argument also holds in the other direction:

$$\inf_{A_0} \|H_1 - A_0 H_2\| \leq \|H_1 - B_1 A_2 H_2\|,$$

$$\leq \|H_1 - B_1 h_1\| + \|B_1 h_1 - B_1 A_2 H_2\|,$$
$$\leq \delta_1 + \|B_1\|_{\text{op}} (\|h_1 - h_2\| + \|h_2 - A_2 H_2\|),$$
$$\leq \delta_1 + \delta_2 + \Delta.$$

By Definition G.1, $d_{\text{repr}}(R_1, R_2) \leq \delta_1 + \delta_2 + \Delta$. □

### G.1 PROOF OF MAIN RESULT 1

For $i \in \{1,2\}$, suppose that $A_i$ is an $\eta_i$-faithful interpretation of $K_i$. Let $\Pi, \Pi_\star$ be alignment classes that map $K_1$'s variables onto $A_1, A_2$'s variables, respectively. Assume that circuits admit $L_i$-layered representations through an alignment map $\rho_i$. Let $F, \tilde{F} \in \Pi^{-1}(A_1)$, then $\mathbb{V}^F : \Sigma^\star \to \mathbb{R}^n$ for some $n \in \mathbb{Z}^+$ (Definition 4.2).

**Assumption G.4.** Let $\Phi : \mathbb{R}^n \to \mathbb{R}^m$ and assume the implementation distance $d$ is in the form

$$d(F, \tilde{F}) \triangleq \|\Phi\mathbb{V}^F - \Phi\mathbb{V}^{\tilde{F}}\|,$$

for some function norm.

$\Phi$ is a **distillation map** that compiles all of the activations of $\mathbb{V}^F$ into a tractable representation. In a sense, $\Phi$ exposes the measurable part of $\mathbb{V}^F$. Denote $R_F$ to be the representations of $F$ constructed through the alignment map $\rho_1$. Since $R_F$ is a deterministic causal model (Definition 4.3), we denote $\mathbb{H}^F : \Sigma^\star \to \mathbb{R}^p$ to be the solution of all variables in $R_F$ given some input setting. In other words, $\mathbb{H}^F$ is the concatenation of all hidden representations of $R_F$. This is the same construction used in Definition G.1.

**Assumption G.5.** Assume there exists a 1-Lipschitz map $\Psi : \mathbb{R}^p \to \mathbb{R}^m$ such that

$$\|\Psi\mathbb{H}^F - \Psi\mathbb{H}^{\tilde{F}}\| \leq d_{\text{repr}}(R_F, R_{\tilde{F}}) \qquad \text{and} \qquad \|\Phi\mathbb{V}^F - \Psi\mathbb{H}^F\| < \omega. \tag{G.5}$$

In other words, the measurable part of our circuit is approximately determined by its representation. Next, we assume that there exists at least one implementation in $\Pi_\star^{-1}(A_2)$ that admits a reasonable representation under $\rho_1$. Since we only enforce that $\rho_1$ is a valid abstraction map for $K_1$, without this assumption, one could construct an adversarial implementation of $A_2$ whose causal variables are completely discarded by $\rho_1$.

**Assumption G.6.** There exists $F^\star \in \Pi_\star^{-1}(A_2)$ such that its representation under $\rho_1$, $R_{F^\star}$, is an $(L_1, \delta_1)$-representation.

We are now ready to prove the main result.

**Main Result.** *Let $\Pi$ and $\Pi_\star$ be alignment classes that map $K_1$'s variables into $A_1, A_2$'s variables, respectively. Then, the approximate interpretive equivalence of $A_1, A_2$ under alignments $\Pi^{-1}, \Pi_\star^{-1}$ is*

$$d_{\text{interp}}(A_1, A_2) \leq \kappa(A_1, K_1, \Pi) + \kappa(A_2, K_1, \Pi_\star) + 2\omega + d_{\text{repr}}(R_1, R_2) + \delta_1 + \delta_2 + \Delta. \tag{G.6}$$

*Proof.* By Lemma G.2, for any $F \in \Pi^{-1}(A_1), \tilde{F} \in \Pi_\star^{-1}(A_2)$,

$$d_{\text{interp}}(A_1, A_2) \leq \kappa(A_1, K_1, \Pi) + \kappa(A_2, K_1, \Pi_\star) + d(F, \tilde{F}).$$

So it remains to upper bound $d(F, \tilde{F})$ for a single pair $(F, \tilde{F})$. Let us choose $F$ to be the circuit $K_1$ which has representations $R_1$. By Assumption G.6, choose $\tilde{F} \in \Pi_\star^{-1}(A_2)$ that admits an $(L_1, \delta_1)$-representation (which we denote as $R_{\tilde{F}}$) under the alignment $\rho_1$. By Assumption G.4,

$$d(F, \tilde{F}) \triangleq \|\Phi\mathbb{V}^F - \Phi\mathbb{V}^{\tilde{F}}\|.$$

We add and subtract $\Psi\mathbb{H}^F, \Psi\mathbb{H}^{\tilde{F}}$,

$$d(F,\tilde{F}) \leq \|\Phi\mathbb{V}^F - \Psi\mathbb{H}^F\| + \|\Psi\mathbb{H}^F - \Psi\mathbb{H}^{\tilde{F}}\| + \|\Psi\mathbb{H}^{\tilde{F}} - \Phi\mathbb{V}^{\tilde{F}}\|,$$

$$\leq 2\omega + \|\Psi\mathbb{H}^F - \Psi\mathbb{H}^{\tilde{F}}\|.$$

Since $\Psi$ is 1-Lipschitz $\|\Psi\mathbb{H}^F - \Psi\mathbb{H}^{\tilde{F}}\| \leq d_{\text{repr}}(R_1, R_{\tilde{F}})$, where $R_{\tilde{F}}$ is the causal model associated with $\mathbb{H}^{\tilde{F}}$. By triangle inequality,

$$d_{\text{repr}}(R_1, R_{\tilde{F}}) \leq d_{\text{repr}}(R_1, R_2) + d_{\text{repr}}(R_2, R_{\tilde{F}}),$$

$$\leq d_{\text{repr}}(R_1, R_2) + \delta_1 + \delta_2 + \Delta.$$

The last inequality follows from Assumption G.6 and Lemma G.3. Putting it all together,

$$d_{\text{interp}}(A_1, A_2) \leq \kappa(A_1, K_1, \Pi) + \kappa(A_2, K_1, \Pi_\star) + 2\omega + d_{\text{repr}}(R_1, R_2) + \delta_1 + \delta_2 + \Delta.$$

$\square$

## G.2 PROOF OF MAIN RESULT 2

For $i \in \{1,2\}$, let us denote $A_i$ as $\eta_i$-faithful interpretations of $K_i$.

**Main Result.** *Let $\Pi$ and $\Pi_\star$ be an alignment that map $K_1$'s variables onto $A_1, A_2$'s variables. Suppose $A_1, A_2$ are $\varepsilon$-approximate interpretively equivalent under the alignment classes $\Pi_\star, \Pi$. Then,*

$$d_{\text{repr}}(R_1, R_2) \leq \varepsilon(\text{Lip}(\rho_1) + 1) + \delta_1 + \delta_2 + \Delta. \tag{G.7}$$

*Proof.* Since $A_1, A_2$ are $\varepsilon$-interpretively equivalent under the alignment cases $\Pi, \Pi_\star$, there exists $F_2' \in \Pi_\star(A_2)$ such that $d(F_1, F_2') \leq \varepsilon$. Let $R_1, R_2'$ be the representations of $F_1, F_2'$ under the alignment $\rho_1$, respectively. Then, $\rho_1$ is Lipschitz continuous implies that

$$d_{\text{repr}}(R_1, R_2') \leq \text{Lip}(\rho_1)d(F_1, F_2') \leq \text{Lip}(\rho_1)\varepsilon. \tag{G.8}$$

Since $R_1$ is an $(L_1, \delta_1)$-representation of $h_1$, there exists $A_1, B_1$ with $\|A_1\|_{\text{op}}, \|B_1\|_{\text{op}} \leq 1$ such that

$$\|A_1 H_1 - h_1\| \leq \delta_1 \qquad \text{and} \qquad \|H_1 - B_1 h_1\| \leq \delta_1,$$

where $H_1$ is the concatenation of all variables in $R_1$. $d_{\text{repr}}(R_1, R_2') \leq \text{Lip}(\rho_1)\varepsilon$ implies there exists $A', B'$ such that

$$\|A'H_1 - H_2'\| \leq \text{Lip}(\rho_1)\varepsilon \qquad \text{and} \qquad \|H_1 - B'H_2'\| \leq \text{Lip}(\rho_1)\varepsilon,$$

where $H_2'$ is the concatenation of all variables in $R_2'$. We now bound the "encoder" and "decoder" error of $R_2'$ separately:

$$\|A_1 A' H_2' - h_2\| \leq \|A_1(A'H_2' - H_1)\| + \|A_1 H_1 - h_1\| + \|h_1 - h_2\|,$$

$$\leq \|A_1\|_{\text{op}}\|A'H_2' - H_1\| + \delta_1 + \Delta,$$

$$\leq \text{Lip}(\rho_1)\varepsilon + \delta_1 + \Delta.$$

And,

$$\|H_2' - B'B_1 h_2\| \leq \|H_2' - B'H_1\| + \|B'(H_1 - B_1 h_1)\| + \|B'B_1(h_1 - h_2)\|,$$

$$\leq \text{Lip}(\rho_1)\varepsilon + \delta_1 + \Delta.$$

Therefore, $R_2'$ is a $(L_1, \text{Lip}(\rho_1)\varepsilon + \delta_1 + \Delta)$-representation of $h_2$. Directly applying Lemma G.3, $d_{\text{repr}}(R_2', R_2) \leq \text{Lip}(\rho_1)\varepsilon + \delta_1 + \delta_2 + \Delta$. By Lemma G.1,

$$d_{\text{repr}}(R_1, R_2) \leq d_{\text{repr}}(R_1, R_2') + d_{\text{repr}}(R_2', R_2) \leq \varepsilon(\text{Lip}(\rho_1) + 1) + \delta_1 + \delta_2 + \Delta.$$

$\square$

**Where is faithfulness in all of this?** Unintuitively, neither Main Result 1 nor 2 explicitly contain the faithfulness of $A_1, A_2$ ($\eta_1, \eta_2$, respectively). Our insight is that faithfulness is implicitly encoded into the alignment classes $\Pi, \Pi_\star$. This is because any valid alignment must be consistent with the faithfulness of $A_1, A_2$ (see Definition 5.1). In this way, $\eta_1, \eta_2$ determine which alignment classes are non-empty. This is why alignment classes form the crux of our constructions, because they describe both structural and behavioral constraints over abstractions.

# H COMPUTING INTERPRETIVE EQUIVALENCE

For $i \in \{1,2\}$, let $A_i$ be an interpretation and $K_i$ be a circuit. Let $\Pi_i$ be an alignment class that maps $K_i$'s variables into $A_i$'s variables. We define probability spaces $(\Pi_i^{-1}(A_i), \cdot, \mathbb{P}_{A_i})$. Suppose $\rho_1, \rho_2$ are alignment maps.

**Definition H.1** (Congruity). Denote $F_1, F_1^\star \overset{\text{i.i.d.}}{\sim} \mathbb{P}_1$, $F_2, F_2^\star \overset{\text{i.i.d.}}{\sim} \mathbb{P}_2$. Let $R_1, R_1^\star, R_2, R_2^\star$ be their representations, respectively, under the alignment maps $\rho_1, \rho_2$.

$$p_1 \triangleq \mathbb{P}[d_{\text{repr}}(R_1, R_1^\star) < d_{\text{repr}}(R_1, R_2)] \qquad p_2 \triangleq \mathbb{P}[d_{\text{repr}}(R_2, R_2^\star) < d_{\text{repr}}(R_2, R_1)], \qquad \text{(H.1)}$$

where $\mathbb{P}$ is the joint distribution over $\Pi_1^{-1}(A_1) \times \Pi_2^{-1}(A_2)$. The **congruity** between $\Pi_1^{-1}(A_1), \Pi_2^{-1}(A_2)$ is $1 - |p_1 + p_2 - 1|$.

In one direction (say $p_1$), we sample two implementations from $\Pi_1^{-1}(A_1)$ and one from $\Pi_2^{-1}(A_2)$. $p_1$ is the probability that the two $A_1$ representations are more similar to each other than either does to the $A_2$ implementation. If $\Pi_1^{-1}(A_1) = \Pi_2^{-1}(A_2)$ and $\mathbb{P}_1 = \mathbb{P}_2$, then $p_1 = p_2 = 1/2$ and congruity is 1. In other words, $A_1, A_2$ are completely *congruous* with respect to their representations.

## H.1 PROOF OF MAIN RESULT 3

For clarity, the result is restated.

**Main Result.** *Let $p$ be the expected value of REPRDIST in Algorithm 1, $F_1, F_1^\star \overset{\text{i.i.d.}}{\sim} \mathbb{P}_1$, and $F_2 \overset{\text{i.i.d.}}{\sim} \mathbb{P}_2$. Let $R_1, R_1^\star, R_2$ be their representations under alignment maps $\rho_1, \rho_2$. Then,*

$$p \geq 1 - \inf_{\varepsilon > 0} \left( \mathbb{P}[d_{\text{repr}}(R_1, R_1^\star) > \kappa(A_1, K_1, \Pi_1) + \varepsilon] + \mathbb{P}[d_{\text{repr}}(R_1, R_2) < d_{\text{interp}}(A_1, A_2) - \varepsilon] \right). \quad \text{(H.2)}$$

*Proof.* Let $R_1, R_1^\star \overset{\text{i.i.d.}}{\sim} \rho_1 \mathbb{P}_1$ and $R_2 \overset{\text{i.i.d.}}{\sim} \rho_2 \mathbb{P}_2$, where $\rho_i \mathbb{P}_i$ is the pushforward of $\mathbb{P}_i$ under $\rho_i$. Consider the event $d_{\text{repr}}(R_1, R_1^\star) > d_{\text{repr}}(R_1, R_2)$. For arbitrary $\varepsilon > 0$, this event occurs with probability 1 when $d_{\text{repr}}(R_1, R_2) > \kappa(A_1, K_1, \Pi_1) + \varepsilon$ and $d_{\text{repr}}(R_1, R_3) < d_{\text{interp}}(A_1, A_2) - \varepsilon$. Therefore,

$$p \geq 1 - \mathbb{P}[d_{\text{repr}}(R_1, R_1^\star) > \kappa(A_1, K_1, \Pi_1) + \varepsilon] - \mathbb{P}[d_{\text{repr}}(R_1, R_2) < d_{\text{interp}}(A_1, A_2) - \varepsilon], \qquad \text{(H.3)}$$

since $\varepsilon > 0$ was arbitrary, we yield the desired result. $\qquad \square$

**Corollary H.1.** *Let $F_1, F_1^\star \overset{\text{i.i.d.}}{\sim} \mathbb{P}_1, F_2, F_2^\star \overset{\text{i.i.d.}}{\sim} \mathbb{P}_2$. Let $\tilde{C}$ be the congruity between $\Pi_1^{-1}(A_1), \Pi_2^{-1}(A_2)$. Define*

$$\begin{aligned}
f(F_1, F_1^\star, F_2, F_2^\star) \triangleq \inf_{\varepsilon > 0} \Big[ &\mathbb{P}[d_{\text{repr}}(R_1, R_1^\star) > \kappa(A_1, K_1, \Pi_1) + \varepsilon] \\
&+ 2\mathbb{P}[d_{\text{repr}}(R_1, R_2) < d_{\text{interp}}(A_1, A_2) - \varepsilon]) \\
&+ (\mathbb{P}[d_{\text{repr}}(R_2, R_2^\star) > \kappa(A_2, K_2, \Pi_2) + \varepsilon]) \Big].
\end{aligned} \qquad \text{(H.4)}$$

*Then,*

$$2 - f(F_1, F_1^\star, F_2, F_2^\star) \leq \tilde{C} \leq f(F_1, F_1^\star, F_2), \qquad \text{(H.5)}$$

*when $f(F_1, F_1^\star, F_2, F_2^\star) \geq 1$.*

*Proof.* By Definition H.1, $\tilde{C}(p_1, p_2) \triangleq 1 - |p_1 + p_2 - 1|$. We break $\tilde{C}$ into two cases:

$$\tilde{C}(p_1, p_2) = \begin{cases} 2 - (p_1 + p_2) & \text{if } 1 \leq p_1 + p_2 \leq 2, \\ p_1 + p_2 & \text{if } 0 \leq p_1 + p_2 \leq 1. \end{cases}$$

Since $\tilde{C}(p_1 + p_2) = \tilde{C}(2 - p_1 - p_2)$, any lower (resp. upper) bound proved on one side transfers to a lower (resp. upper) bound on the reflected side. Thus, we separately bound each case. Consider first $1 \leq p_1 + p_2 \leq 2$. Then,

$$\tilde{C}(p_1, p_2) = 2 - (p_1 + p_2),$$

$$\leq \inf_{\varepsilon > 0} \Big[ \mathbb{P}[d_{\text{repr}}(R_1, R_1^\star) > \kappa(A_1, K_1, \Pi_1) + \varepsilon]$$

$$+ 2\mathbb{P}[d_{\text{repr}}(R_1, R_2) < d_{\text{interp}}(A_1, A_2) - \varepsilon]) + (\mathbb{P}[d_{\text{repr}}(R_2, R_2^\star) > \kappa(A_2, K_2, \Pi_2) + \varepsilon]) \Big],$$

where the last inequality follows from Equation (H.2). Now the case $0 \leq p_1 + p_2 \leq 1$,

$$\tilde{C}(p_1, p_2) = p_1 + p_2,$$

$$\geq 2 - \inf_{\varepsilon > 0} \Big[ \mathbb{P}[d_{\text{repr}}(R_1, R_1^\star) > \kappa(A_1, K_1, \Pi_1) + \varepsilon]$$

$$+ 2\mathbb{P}[d_{\text{repr}}(R_1, R_2) < d_{\text{interp}}(A_1, A_2) - \varepsilon]) + (\mathbb{P}[d_{\text{repr}}(R_2, R_2^\star) > \kappa(A_2, K_2, \Pi_2) + \varepsilon]) \Big].$$

$\square$

## H.2 Intervention-Implementation Duality

Our results in the previous sections show that estimating both the representation similarity and interpretive compression is crucial to understanding interpretive equivalence. In general, enumerating the set of implementations for a given interpretation is hard. Herein, we make a connection between implementations and interventions. Specifically, we show that it is possible to view interventions as instantiations of new implementations. With this perspective, we bound the sample complexity (in interventions) of interpretive compression estimation.

First, given a set of interventions, we define a notion of closure for alignment classes.

**Definition H.2.** Let $K \triangleq (\mathbb{V}, U, \mathbb{F}, \succ)$ be some circuit and $A$ be an $\eta$-faithful interpretation of $K$. Also, let $\Pi$ be an alignment class that maps $K$'s variables onto $A$'s. Consider a set of interventions over all variables in $\mathbb{V}$ which we denote as $\mathscr{I}(\mathbb{V})$ (see Definition D.2). We say that $\Pi$ is **closed** under $\mathscr{I}(\mathbb{V})$ if for each implementation $K^\star \in \Pi^{-1}(A)$, there exists $\tilde{f} \in \mathscr{I}(\mathbb{V})$ such that $K^\star = \text{do}(\mathbb{V} \leftarrow \tilde{f})$.

*Remark.* Any alignment class $\Pi$ is closed under a **natural set of interventions**. Simply, given circuits $K, K^\star \in \Pi^{-1}(A)$, this is the set of interventions that replace $K$'s transition functions, $\mathbb{F}$, with *all* of $K^\star$'s transition functions, $\mathbb{F}^\star$.

*Remark.* Dually, instead of explicitly enumerating an alignment class, we could instead specify a set of interventions under which $K$'s interpretation is preserved. This implicitly induces an alignment class. With this method, one only needs to argue that the interventions are interpretation-preserving. This is the method we use in GETIMPL to construct implementation sets.

**Definition H.3.** Let $C$ be some set and $d : C \times C \to \mathbb{R}^{\geq 0}$ be some distance metric over $C$. Let $\{x_1, \ldots, x_m\} \subset C$. Then, the **empirical diameter** of $C$ is $\max_{1 \leq i < j \leq m} d(x_i, x_j)$.

**Proposition H.2.** *Let $d : \mathbb{V} \times \mathbb{V} \to \mathbb{R}^{\geq 0}$ be a distance metric between circuits. Let $K$ be a circuit, $A$ be some $\eta$-interpretation of $K \triangleq (\mathbb{V}, U, \mathbb{F}, \succ)$, and $\Pi$ be an alignment class. Let $\mathscr{I}(\mathbb{V})$ be a set of interventions on $K$ under which $\Pi$ is closed. Let $\mathbb{P}$ be a distribution over $\mathscr{I}$. Let $C = N(\Pi^{-1}(A), d, \varepsilon/2)$ be the covering number of the implementation space $\Pi^{-1}(A)$. Define*

$$p_{\min} \triangleq \min_{1 \leq j \leq C} \mathbb{P}[\{\iota \in \mathscr{I}(\mathbb{V}) : \text{do}(\mathbb{V} \leftarrow \iota) \in B_j\}],$$

*where $\{B_i\}$ are balls of radius $\varepsilon/2$ that cover $\Pi^{-1}(A)$. Then, for*

$$m \geq \frac{\log(C) - \ln(\delta)}{p_{\min}},$$

*we have that $\mathbb{P}[\kappa(A, K, \Pi) - \hat{\kappa}_m(A, K, \Pi) \leq \varepsilon] > 1 - \delta$ where $\hat{\kappa}$ is the empirical diameter of $\Pi^{-1}(A)$ computed from $m$ sampled interventions.*

*Proof.* From Definition H.3, $\mathbb{P}[\kappa(A, K, \Pi) \geq \hat{\kappa}_m(A, K, \Pi)] = 1$. Consider the event $\kappa(A, K, \Pi) - \hat{\kappa}(A, K, \Pi) > \varepsilon$. This occurs only if we draw $m$ interventions $\{\iota_1, \ldots, \iota_m\}$ where there exists at least one ball $B_i$ where there does not exist some $\iota_i$ such that $\text{do}(\mathbb{V} \leftarrow \iota_i) \in B_i$. This event occurs with probability at most $(1 - p_{\min})^m$. By the Poisson approximation, we have that $(1 - p_{\min})^m \leq e^{-mp_{\min}}$. Setting this less than $\delta$ and rearranging, we yield the result. $\square$

