# OpenReview forum: "Tracking Equivalent Mechanistic Interpretations Across Neural Networks"
_ICLR.cc/2026/Conference — ICLR 2026 Poster_

### Official Review · Reviewer_DWxu · 2025-10-29

**Soundness:** 3
**Presentation:** 4
**Contribution:** 4
**Rating:** 8
**Confidence:** 3

**Summary:**

- This paper studies the problem of “interpretive equivalence”, which is how to determine whether two models share the same interpretation of a behavior (and by extension implementation). They develop algorithms to measure the similarity of two interpretations of a model based on their representations, and evaluate their algorithms/theory on three tasks using both toy transformers and pretrained language models.
____
- The paper seems like a solid contribution towards making the evaluation of neural network interpretations more rigorous, and for evaluating the equivalence of interpretations across models.

**Strengths:**

- The paper is well written and its connection to previous work is evidently thought-through throughout the paper.
- There is a nice mix of theory and practice - providing intuition and showing the framework being applied to both toy settings with known ground truth and slightly more complex settings that have some level of prior understanding.
- The evidence that this framework can distinguish between implementations on the toy task n-Permutation setting is promising (Section 3.1, Figure 2(left)).

**Weaknesses:**

- The argument that interpretive equivalence allows us to study simpler/smaller models (Line 90) seems like a nice implication of this framework when the ambiguity between two models is high. However, I have two concerns with this:
    - The first is that it seems desirable that ambiguity between the same model usually close to 1 for the same task, but it sometimes is not (e.g. GPT2-M w/ itself on IOI task). Does this mean the perturbation of GETIMPL is too aggressive, or what else might cause this?
    - Second, this is somewhat at odds with evidence from other work that some interesting capabilities seem to appear only in larger models (see e.g. [Brown et al], [Wei et al], [Schaeffer et al]). A recent example that comes to mind from the mechanistic interpretability literature is [Prakash et al] who note “we do not examine smaller models, as they are unable to coherently solve the … task”. How does your framework incorporate how well a model can solve a task? Do you have some performance threshold you decide before you choose to use its representations to compare to another model? While the tasks analyzed here are simple and I’d expect all the models you analyze can do them, it's unclear how model performance is factored in when understanding the similarity of implementation for more complex tasks, which seems like a current limitation of the proposed method.

- The GETIMPL procedure in Algorithm 1 lacks some preciseness that would help clarify my understanding of the Ambiguity computation. As I understand it, the GETIMPL is designed to produce a modified model with an equivalent implementation/interpretation. However, what gets returned seems to be the exact copy of the model you start with, rather than a modified version. It would be helpful to clarify whether the operations in GETIMPL are in fact supposed to be producing a modified copy of the model (with an equivalent interpretation/implementation), or whether it does actual return the same $h_{\theta}$ that is provided as an input.

- It's unclear to me how to interpret a middle value of "ambiguity". Does this mean that _some_ of the implementation is similar? What exactly does that even mean? While not having needing a specific interpretation to base the metric in can be a strength of the method in some ways (e.g. when ambiguity is high, or prior work has come up with "similar" interpretations, confidence can be high that the ambiguity score is meaningful) it seems like this can also be a limitation as without an interpretation, an ambiguity of "0.4" seems hard to get much out of. I could imagine applying targeted version of this same algorithm to validate specific functional roles of various components (measuring whether "mover heads" perform similar roles, or "induction heads" act similarly across models, etc.), but this requires a prior interpretation.

___
- Brown et al. [Language models are few-shot learners](https://papers.nips.cc/paper/2020/hash/1457c0d6bfcb4967418bfb8ac142f64a-Abstract.html)

- Wei et al. [Emergent Abilities of Large Language Models](https://openreview.net/forum?id=yzkSU5zdwD)

- Schaffer et al [Are Emergent Abilities of Large Language Models a Mirage?](https://openreview.net/forum?id=ITw9edRDlD)

- Prakash et al. [Language Models use Lookbacks to Track Beliefs](https://arxiv.org/abs/2505.14685)

**Questions:**

Overall, I think the paper is pretty good. My main concerns are listed in the "weaknesses" section, and addressing those would help me feel more confident in the paper. Below, I share minor comments/questions.

- I’m not sure what the text at the top of Figure 1 is trying to communicate, can you elaborate?
- Line 284: When you say “GPT2” is this GPT2-xl or GPT2-small?
- Line 281: “interpretive similar” seems to be a new term. Is it just a weaker form of interpretive equivalence? It feels natural, but you might want to define it somewhere.
- It’s kind of surprising that the diagonal for Figure 2b is not higher than some off-diagonals (particularly for the gpt2 family) - any ideas what's going on there?
- Another further validation of this method might be to compute ambiguity for individual components with specific hypothesized roles, like induction heads. Have you tested whether your ambiguity score yields high values when only measuring it on component subsets (like induction heads on repeated text snippets)?

Minor Typos:
- Line 215 (footnote): “the two models do not the same interpretation” -> missing “share” (ie. do not __share__ the same…)
- Line 357: “abstrations” -> abstractions
- Line 411: “On the other,” -> On the other hand?
- Line 423: “simiarlity” -> similarity
- Line 946: “framemworks” -> frameworks

Papers:
- You may also be interested in [Miller et al.]’s analysis of evaluating and comparing circuits, which seems related to the problem you’re proposing and addressing here.

___
Miller et al. [Transformer Circuit Metrics are Not Robust](https://openreview.net/forum?id=zSf8PJyQb2)

---

> ### Author Response · Authors · 2025-11-20
>
> We thank the reviewer for their time and detailed feedback. We are grateful that the reviewer recognized the mix of contributions of our paper across theory and practice. We hope to address some of your concerns below and look forward to further engaging with you during this discussion period.
>
> > …seems desirable that ambiguity…close to 1 for the same task, but it sometimes is not…Figure 2(left)...any ideas what's going on there?
>
> Yes! We want maximum ambiguity when comparing a model with itself. It is surprising that this does not occur in Figure 2(left). We provide two hypotheses for this phenomenon based on the geometry of the sets of implementations:
>
> 1.	This could be due to sampling noise. Concretely, this discrepancy could be driven by the interpretive compression of GPT2. Large interpretive compression of the GPT2-family implies that there are many implementations of GPT2-small/medium’s IOI interpretations. This would then require more sampled implementations to completely identify each of them.
> 2.	It may also have to do with the geometry of the implementation space. For example, the geometry of GPT2-small and -medium's implementation space separately may be very pathological (i.e. has non-smooth boundary). However, the union of their implementation space may be geometrically nice (see Figure 3). In turn, this results in a much smaller sample complexity when we apply Algorithm 1 in across both implementations but higher sample complexity when Algorithm 1 is applied separately.
>
> The resolution of these hypotheses is important to pin down the stability of Algorithm 1 but may require significantly more analysis into the implementation sets themselves. Thus, we leave this open for future work. We have added this discussion to Appendix A.
>
>
> > …at odds with evidence from other work that some interesting capabilities seem to appear only in larger models…How does your framework incorporate how well a model can solve a task?
>
> This is a great point. Our framework does capture this implicitly through two mechanisms:
>
> 1. Assumption 2 in Section 6. We require the two models we compare to be functionally similar (Lines 437-438; 442-443). This is consistent with the statement by [Prakash et al. 2023](https://arxiv.org/abs/2505.14685) because if the large model adheres to ground truth and the smaller model cannot perform equally well, then simply by virtue of triangle inequality their functional behavior cannot be similar.
>
> Conversely though our framework introduces the following interesting possibility: take a small model that does not perform well but **fails in a similar way as this large model**, restrict our subtask S to be these failure cases. Maybe we could diagnose why the large model is failing by interpreting the failure cases of the small model?
>
> 2. Interpretive compression. Notice that our bounds in Main Result 1 & 2 depend strongly on this quantity. Consider a model that is $\eta$-faithful (i.e. its interpretation performs badly), as $\eta$ increases the set of implementations of this "unfaithful" interpretation grows monotonically. This implies that interpretive compression increases monotonically. By Main Result 1 (Lines 451-455), representation similarity between these sets of implementations tells us virtually nothing about their interpretive equivalence.
>
> Our framework makes clear that we should compare functionally similar models. This has nothing to do with the size of the model per se. We see the ideal use case of Algorithm 1 as applying Example 1.1 and Example 1.2 in conjunction: **first decompose a large model on a complex task into the same model on interpretive equivalent subtasks. Then, use smaller models to approximate each of these much simpler tasks.** Results in Figure 2(b,c) show that each of these steps are separately possible.
>
> > …performance threshold…before you choose to use its representations to compare to another model?
>
> For permutation detection and IOI, we enforce that our models perform the task almost perfectly. This is because we are asking the question: given the task=ground truth prediction, how similar are model mechanism? In the case of permutation detection all implementations achieve >96% accuracy on the task. While for IOI all models can perform the task with >98% accuracy. With respect to choosing the representations, for permutation detection and IOI we use all representations from all hidden layers.
>
> In contrast for POS, we are enforcing a different condition: given that the task=model’s own behavior, how similar are the model mechanisms? Thus, in this case, the model’s performance relative to itself is always 100%. With respect to choosing the representations, we use a function vector approach detailed in Appendix E.5 and choose representations from attention heads that contribute positively to model behavior (i.e. after we remove this attention head, model behavior shows non-zero deviation from baseline; Lines 1230-1241).

---

> > ### Author Response · Authors · 2025-11-20
> >
> > > …unclear how model performance is factored in when understanding the similarity of implementation for more complex tasks, which seems like a current limitation of the proposed method.
> >
> > We acknowledge that this is a limitation. Practically, we only interpret models that show good performance. However, we argue that this is a standard convention in the mechanistic interpretability community (leveraging your same quotation above from [Prakash et al. 2023](https://arxiv.org/abs/2505.14685) :) also [Wang et al. 2023](https://arxiv.org/abs/2211.00593); [Shi et al. 2024](https://arxiv.org/abs/2410.13032)). As discussed above, one possible extension is we can use this framework to measure failure cases as well. This is possible because we do not necessarily require that model performance is good with respect to ground truth only that the two models we are comparing are behaviorally similar (again, re. Assumption 2, Section 6).
> >
> > > It would be helpful to clarify whether the operations in GETIMPL are in fact supposed to be producing a modified copy of the model (with an equivalent interpretation/implementation), or whether it does actual return the same that is provided as an input.
> >
> > GetImpl returns a modified model. Take IOI as an example, GetImpl returns the same model but with some of its components “ablated” (every time we pass this model an input, we intervene on these “ablated” components with activations had it been run on a different input).
> >
> > > …how to interpret a middle value of "ambiguity"…What exactly does that even mean?
> >
> > We formally define Ambiguity in Appendix H. Concretely, if $M_1, M_2$ are two models, then it is the probability that representations $R_1, R_2 \sim \mathrm{ImplementationsOf}(M_1)$ no closer than representations $R_1 \sim \mathrm{ImplementationsOf}(M_1), R_2 \sim \mathrm{ImplementationsOf}(M_2)$. Main Result 3 gives some insight into graded notions of ambiguity. Ambiguity is an upper bound on the probability that two models’ representation similarity upperbounds interpretive equivalence. In other words, ambiguity is proportional to how confident a model’s representations similarity upperbounds their interpretive equivalence.
> >
> > > While not having needing a specific interpretation to base the metric in can be a strength of the method in some ways…it seems like this can also be a limitation as without an interpretation, an ambiguity of "0.4" seems hard to get much out of. I could imagine applying targeted version of this same algorithm to validate specific functional roles of various components…but this requires a prior interpretation.
> >
> > You raise an important point here and we acknowledge this as one of the limitations of our method (see Lines 85-87; Appendix A, Lines 814-820). We do not claim that our method necessarily “solves” mechanistic interpretability, but rather this relaxed problem may give practitioners a starting point to interpret and deal larger models. For example, one way to use the graded notion of Ambiguity is analogous to optimizing false discovery rate in hypothesis testing ([Benjamini and Hochberg 1995](https://mathscinet.ams.org/mathscinet/relay-station?mr=1325392)). A practitioner a priori set some ambiguity threshold, compare a target large model of study with a host of smaller ones, and then selectively prunes out small models that do not align with this large model. The point is that ambiguity allow practitioners an automated way to quickly reduce the search space of possible interpretation hypotheses without doing a mechanistic interpretation. But we agree that to truly understand what is going on inside of these models’ interpretive equivalence on its own is not enough.
> >
> > While this falls outside the scope of our work, we invite the reviewer to consider use cases of ambiguity that may extend beyond interpretability where concrete interpretations may not be needed. For example, [Lee et al. 2024](https://arxiv.org/abs/2401.01967) show that alignment algorithms mechanistically change the model. Practitioners could apply interpretive equivalence to models before and after post-training alignment to quickly quantify the efficacy of their treatment on the actual mechanism of the model. This type of evaluation could be more robust to out-of-distribution evaluation as seen in [He et al. 2024](https://arxiv.org/abs/2406.02550).
> >
> > > I’m not sure what the text at the top of Figure 1 is trying to communicate, can you elaborate?
> >
> > From left to right, we are trying to express that interpretations come from causal abstraction of circuits, and circuits can be thought of as causal implementations of representations. We welcome any feedback to make these relationships clearer.
> >
> > > Line 284: When you say “GPT2” is this GPT2-xl or GPT2-small?
> >
> > We are using GPT2-small, we are using these pretrained weights from [HuggingFace](https://huggingface.co/openai-community/gpt2). We have clarified in the uploaded revision. Please see Line 283-284.

---

> ### Author Response · Authors · 2025-11-20
>
> > …“interpretive similar”…you might want to define it somewhere.
>
> At this point in the manuscript, we mean this informally in the sense that if we perform apply the procedure of mechanistic interpretation onto both networks, we should see algorithms that are heuristically similar to each other. We formally define “interpretive similarity” in Lines 401-403. We have updated the manuscript with a link to that section.
>
> > Another further validation of this method might be to compute ambiguity for individual components with specific hypothesized roles, like induction heads. Have you tested whether your ambiguity score yields high values when only measuring it on component subsets (like induction heads on repeated text snippets)?
>
> In short, we have not tested our method on such fine-grained components. However, the IOI task is heavily reliant on two components: induction heads and copy suppression heads in which our method does quite well in. With respect to induction heads, our understanding is that an induction head refers to both an interpretation but also an implementation, it’s unclear to us how to test interpretive equivalence in a nontrivial manner here, we would appreciate some clarifications here, or if you have any ideas!
>
> > Typos
>
> We appreciate your close proofing; we have corrected these typos in the uploaded revision. We have indicated these points of revision with "FIX" in the right margin. Thank you!
>
>
> **We apologize for the long rebuttal and thank the reviewer for their time, patience, and detailed feedback.** You have raised many interesting points that have made us think very deeply about the implications of our work and we want to make sure that we completely address your concerns. We are excited to further engage with your during this discussion period.
>
> Best,
>
> Authors

---

> > ### Comment · Reviewer_DWxu · 2025-11-24
> >
> > Thank you for the detailed response to my questions and various clarifications. The updates have made the paper stronger, and the limitations notes are helpful. I appreciated the note around uses cases beyond interpretability. I enjoyed this paper and think it will be a nice contribution to the conference. I plan on maintaining my positive score.

---

### Official Review · Reviewer_pr5o · 2025-10-31

**Soundness:** 3
**Presentation:** 4
**Contribution:** 2
**Rating:** 6
**Confidence:** 3

**Summary:**

The paper introduces a novel framework for determining interpretive equivalence between neural networks, whether two models implement the same mechanistic interpretation without requiring explicit interpretation of either model.
The core insight is nice and sidesteps identifiability difficulties in MI by solving a relaxed problem: two interpretations are equivalent if and only if the set of possible implementations are similar. The authors operationalize this through Algorithm 1 (AMBIGUITY), which leverages representation similarity across sampled implementations to estimate equivalence.

**Strengths:**

Novel problem formulation: The shift from "interpret each model" to "determine if models share interpretations" is valuable and practical as a conceptual idea.
Theoretical grounding: Main Results 1-3 provide some theoretical guarantees via lower and upper-bounds connecting representational similarity with interpretative equivalence.
The advantage of the algorithm 1 is that it is scalable and applied to some standard LLMs.
I appreciate that the authors put effort in the presentation of the work and the concepts.

**Weaknesses:**

Despite the framework being stated in very general terms with causal abstraction as its main language, it seems that Algorithm 1 is fairly tied to circuit (with GetImpl doing circuit manipulations basically).

The paper acknowledges that the linearity assumption is used and could be studied further in future work but this remains a fairly strong and important assumption.
In particular, the findings of Figure 2 do not really prove the usefulness of the approach, to produce the same kind of heatmaps it is enough to distinguish model class (on the left panel) or model family (on the center panel), which intuitively is easy to do by looking at their representation, and most distances between representations would produce such heatmaps without making claims about interpretative equivalence. I think it would be more convincing to show cases where "standard (CCA-like)" representational similarity do not give the correct answer, and AMBIGUITY does. The POS tagging experiment is already a bit more convincing as it already compare against an alternative hypothesis.

I fear it might be easy to engineer counter-examples for Algorithm 1:
- false positives: It would be two neural networks whose representations are close to be linearly mappable to each other but who do different things (maybe even beyond different implementation of the same behavior, different behavior altogether). Experiment idea: Start with a network and task, train another neural network adversarially to behave maximally differently from the first one but constrained or regularized to enforce linearly mappable representations. Failure to produce such adversarial examples would be a strong demonstration of Algorithm 1's robustness to false positive.
- false negatives: It would be two neural network who do implement the same thing but it is not visible by linear inspection of their representations. Experiment Idea: if you apply invertible non-linear transforms at each layer of network to produce a new network, linearity is lost but functionality is unchanged.

Convergence and stability of Algorithm 1:
I think the paper could engage more with the convergence of algorithm 1, it depends on finite sampling of the number of repeats n in AMBIGUITY. It would be nice to see convergence rates and stability (as measured by some notion of variance of the estimate)

**Questions:**

What are the convergence properties of Algorithm 1?
How would standard representation comparison methods fare for the tasks in Figure 2.

---

> ### Author Response · Authors · 2025-11-20
>
> We thank the reviewer for their time, valuable feedback and criticisms of our work. Below, we hope to address some of your concerns and provide points of clarification. We very much look forward to further engaging with you during this discussion and revision period.
>
> > Despite the framework being stated in very general terms with causal abstraction as its main language, it seems that Algorithm 1 is fairly tied to circuit (with GetImpl doing circuit manipulations basically).
>
> You are right. Algorithm 1 is wholly tied to the model’s circuit. Our treatment of circuits, however, is different than the desiderata of circuitry seen in mechanistic interpretability. Most notably, we **do not** assume minimality (see Definition B.2). In this way, a circuit is just the model’s computational graph. This relaxation also leads to significant computational benefits for circuit discovery see [Aldofi, Vilas, and Wareham (2025)](https://arxiv.org/abs/2410.08025).
>
> This notion of a circuit is necessary to describe an interpretation, as an interpretation needs to abstract some underlying process (and the circuit rigorously describes what this process is). In this way, our framework does not lose generality.
>
> > linearity assumption is used and could be studied further in future work but this remains a fairly strong and important assumption.
>
> Algorithm 1 is not dependent on any specific notion of representation similarity. If d_repr has the properties of a pseudo-metric, our formalisms of Ambiguity still hold along with Main Result 3 and Appendix H.
>
> First, the linearity assumption we use is well-studied in existing mechanistic interpretability work and beyond (see linear representation hypothesis; [Alain and Bengio 2018](https://arxiv.org/pdf/1610.01644), [Belinkov 2022](https://direct.mit.edu/coli/article/48/1/207/107571/Probing-Classifiers-Promises-Shortcomings-and); [Roeder, Metz, and Kingma 2021](https://proceedings.mlr.press/v139/roeder21a/roeder21a.pdf); [Park, Choe, and Veitch 2024](https://arxiv.org/pdf/2311.03658)). Thus, although strong, our assumptions are well-founded on empirical results in the literature.
>
> As for the rest of our theory, we acknowledge this limitation and discuss the possibility of using more expressive notions of representation similarity in Lines 451-461.
>
> > In particular, the findings of Figure 2 do not really prove the usefulness of the approach, to produce the same kind of heatmaps it is enough to distinguish model class (on the left panel) or model family (on the center panel), which intuitively is easy to do by looking at their representation, and most distances between representations would produce such heatmaps without making claims about interpretative equivalence. I think it would be more convincing to show cases where "standard (CCA-like)" representational similarity do not give the correct answer, and AMBIGUITY does...
>
> We would like to clarify that our work differs and address several limitations of “standard (CCA-like” representational similarity metrics. We summarize this through two points:
>
> 1.	Representation alignment on its own may not be well-calibrated. For example, graded notions of Ambiguity directly correspond to a probabilistic notion of interpretive equivalence (Main Result 3, Appendix H). This is not true for representation similarity. For example, even for CCA (where it lies between 0-1), to use this for interpretive equivalence, one would first need to calibrate this 0-1 score to a probability of interpretive equivalence. This requires access to the compared model’s interpretations which is what we do not want to assume.
> 2.	As you succinctly describe in your counterexamples below, “standard (CCA-like)” representation similarity metrics are neither necessary nor sufficient to capture interpretive equivalence. For simplicity, suppose that we have a one layer neural network $x\to f_1(x) \to g_1(f_1(x))$ with representations $f_1(x)$ another neural network $x\to f_2(x)\to g_2(f_2(x))$:
>
> **a.** Even if these networks have nothing to do with each other one could modify $f_2$ such that $x \to f_2(x)\to f_2'(f_2(x)) \to g_2(f^{-1'}_2(f_2(x)))$  such that $f’_2$ maximizes similarity with $f_1$.
>
> **b.** Likewise, we can also construct f_2’ to maximize its dissimilarity with f_1, even if the two neural networks are related.
>
> Regarding the experiments in the left panel, each interpretation here is not identified by its “model class” (i.e. its architecture, embedding schema, etc.). That is, for each interpretation, we generate many different implementations of varying architectures.
> Could you clarify what you mean by model class here? This may help us better address your concern.
>
> For the experiment of reducing complex models, we acknowledge that design-grounded representation similarity metrics may be able to distinguish model architectures, but this is unrelated to what we are trying to show here. The point is more so that Ambiguity is invariant with respect to scale.

---

> ### Author Response · Authors · 2025-11-20
>
> > I fear it might be easy to engineer counter-examples for Algorithm 1
>
> We appreciate the reviewer’s insight here. We are that these counterexamples that you raise are precisely the examples that **motivate and differ** our method from “standard (CCA-like)” methods.
>
> **Crucially, we are not simply comparing two neural networks pointwise.** Existing work by [Meloux et al. 2025](https://arxiv.org/abs/2502.20914) explicitly demonstrate that such pointwise comparisons are vacuous. Instead, the novelty of our approach comes in two-fold:
>
> 1.	Instead of comparing a network’s representation with another through representation similarity and inferring their interpretive equivalence through this one comparison, **we are generating implementations for both networks and computing pairwise representation similarity across all these implementations. **
>
> 2.	The problem then is how do we generate these implementations in a tractable way? Our insight here is that for a given model, its implementations can be identified with a dual space of interventions from our base model. This is what gives rise to Algorithm 1.
>
> The counterexamples that you construct is essentially what we are visualizing in Figure 3.
>
> That is, we could sample with some non-zero probability two models that lie in the overlap that have high representation similarity but averaged across sampling over the entire space of the geometry of these regions and their differences become clear. This would address your concern about **false positives**.
>
> On the other hand, say we have two models the exact same implementation space $S$. There again is some non-zero probability that we sample two models $h, h’ \in S$ such that $h \neq h’$. But, averaged over samples over the entirety of S, our derivation in Lines 152-160 hold and we see an Ambiguity of 1. This would address your concern about **false negatives**.
>
> **Our insight here is that interventions drive this identification process not the representation similarity metric.**
>
>
> > What are the convergence properties of Algorithm 1?
>
> We derive the sample complexity of Algorithm 1 in Appendix H. Concretely, it is proportional to the log of the covering number of the implementation space. One way to reason about Algorithm 1 is that it is essentially an MCMC sampling procedure over the uniform distribution of implementations where Steps 2-6 are our proposal mechanisms and Step 7 is our acceptance criterion.
>
> We thank the reviewer for their critical engagement with our work. We hope that we have further clarified our work and look forward to resolving any further concerns or imprecisions that you made find during this discussion period.
>
> Best,
>
> Authors

---

> > ### Comment · Reviewer_pr5o · 2025-11-24
> >
> > Thank you for your detailed answer and clarifications.
> >
> > ### Regarding figure 2:
> > My comment is not a criticism of AMBIGUITY and I do agree with the authors about the limitations of representational similarities but my comment is more a question about whether this Figure actually shows the strength of AMBIGUITY.
> > For the the second panel, the validity of AMBIGUITY is measured by the proxy of its ability to distinguish "actual interpretative differences observed by [previous works]", which turns out to be distinguishing model family, a particularly easy task. I understand that distinguishing model family is not the goal of AMBIGUITY, but an alternative method that would only recognize model family would pass this test. Concerning the left panel, I agree that it is more directly targetted at evaluating AMBIGUITY, but it also fairly narrow-scope because the transformers are obtained by RASP + transformation.
> >
> > ### Regarding counter-examples:
> > > That is, we could sample with some non-zero probability two models that lie in the overlap that have high representation similarity but averaged across sampling over the entire space of the geometry of these regions and their differences become clear. This would address your concern about false positives.
> >
> > > On the other hand, say we have two models the exact same implementation space $S$. There again is some non-zero probability that we sample two models $h, h’ \in S$ such that $h \neq h’$. But, averaged over samples over the entirety of S, our derivation in Lines 152-160 hold and we see an Ambiguity of 1. This would address your concern about false negatives.
> >
> > I would agree with this but this is not demonstrated empirically.
> >
> >
> > Despite these comments I still emphasize that I do find the work valuable do think AMBIGUITY is a solid contribution which is reflected by my positive assessment of the work and I'd be completely satisfied to see this paper published at ICLR.
> > I just think the work could better demonstrate empirically its validity. Currently the evaluation consists in the two fairly narrow-scope synthetic experiments of Figure 2 coupled with the theoretical justification. I feel that demonstrating practical robustness of AMBIGUITY to adversarial scenarios which, as you discussed, can be constructed from the Figure 3 and demonstrating empirical convergence properties (the theoretical log n in appendix H is nice to have but this number could itself be very large for practical purposes) would be elevating this paper further.

---

### Official Review · Reviewer_Td1h · 2025-11-01

**Soundness:** 3
**Presentation:** 2
**Contribution:** 2
**Rating:** 4
**Confidence:** 4

**Summary:**

The aim of mechanistic interpretability (MI) is to discover the algorithms that are implemented by trained neural networks to perform tasks. This is a difficult problem, and one of the difficulties is that it is unclear what the right target of MI is: it is relatively rare that behaviour can be captured by “circuits” whose function is completely understood, if by “circuit” one means neurons and edges between them. Indeed many of the most promising approaches to MI now replace neurons by other degrees of freedom, such as SAE features.

The authors of the present paper propose a way of sidestepping this issue by trying to quantify when two neural networks compute in the same way, rather than trying to directly characterise how they compute. This is an interesting idea, which can be viewed as an evolution of the existing literature on representational similarity across models (see e.g. Klabunde et al “Similarity of Neural Network Models: A Survey of Functional and Representational Measures”).

The contribution of the paper comes in two parts: a theoretical part and an empirical part. In the theoretical part the authors propose precise definitions of various terms such as representation, interpretation, circuit and alignment, and prove some simple facts about their key measure of “ambiguity” and representation distance. In the empirical part they test their notion of ambiguity on three main settings: RASP programs, the IOI circuit in GPT2 and Pythia models, and GPT2 between different tokens.

**Strengths:**

* The idea of quantifying representational similarity across a circuit, and the associated metric of ambiguity, is an interesting approach that sidesteps some of the intractable parts of MI
* The paper makes a serious attempt to ground the somewhat informal terms used in MI in the precise language of casual graphs
* Includes an interesting baseline in RASP programs and applications to models up to 2.8B parameters, showing the feasibility of applying the method at scale
* There is a significant amount of new terminology in this paper, but the authors do provide a detailed glossary which I appreciated

**Weaknesses:**

Minor weaknesses:

* I think the related work is not sufficient; in particular, there are not enough references to the existing literature on representational similarity across models which this paper seems to be extending in various ways. Maybe the authors could take a look at Klabunde et al “Similarity of Neural Network Models: A Survey of Functional and Representational Measures”.
* I found the leftmost plot in Fig 2 to be a bit misleading, in the sense that the color coding here seems to refer to some threshold that is not specified rather than e.g. linearly representing the ambiguity as in the middle plot.
* In several places, including (5.1), distance is defined between sets of variables where it seems that what is intended is some average over distances on actual inputs.

Major weaknesses:

* I am not convinced of the importance of the theoretical component of this paper. Could the authors elaborate more on the relation between the empirical and theoretical contributions?
* In my view the main empirical contribution is the middle plot of Fig 2\. However I do not have enough details to evaluate how significant I find these ambiguity figures. At first glance the high ambiguity between Pythia models is interesting, but what are the precise circuits being used here? What are the ablations that are performed? It seems that the meaning of the ReprDist metric depends quite heavily on the details and strength of the ablation and I did not find these details.

**Questions:**

* 260: what are these interventions?
* 288: what does “yield the same interpretation” mean?
* 156: we sample implementations, but according to what distribution? It seems in practice what we do is just what Algorithm 1 says?

---

> ### Author Response · Authors · 2025-11-20
>
> We thank the reviewer for their time and valuable feedback. We hear your concerns and hope that we can thoroughly address them below. We look forward to further engaging with you during this discussion period.
>
> > related work is not sufficient…not enough references to the existing literature on representational similarity across models which this paper seems to be extending in various ways
>
> We appreciate the reviewer’s feedback here. As such, we have added the work of [Klabunde et al. (2025)](https://arxiv.org/pdf/2305.06329) in our related literature (see Appendix D). However, we want to emphasize that our work is fundamentally different than the endeavor of measuring representation similarity in two important ways:
>
> 1.	We are operating at the level of interpretations (i.e. comparing sets of models across their data distributions under interventions rather than performing pairwise comparisons of static models). This distinction is important because representations of individual models are neither necessary nor sufficient to identify interpretations as Reviewer `pr5o` succinctly describes. To summarize using the vocabulary in [Klabunde et al. 2025a](https://arxiv.org/pdf/2305.06329); [2025b](https://arxiv.org/pdf/2408.00531):
> Consider any representation similarity metric designed around prediction and/or design grounding.
>
>          a. There exist two models that share the same prediction and design (like architecture) grounding but have completely different interpretations. Thus, any representation similarity metric designed cannot sufficiently distinguish them.
>
>          b. There exist two models that have slightly different prediction (like Figure 2c, where we have two completely different tasks) and design grounding but share similar interpretations. In this case, any representation similarity is not necessary to group them.
>
> 2.	Representation similarity in our paper is to simply provide a notion of distance between implementations, it is neither necessary nor sufficient to describe what interpretations or implementations are. **Our insight is that representations are useful to describe interpretations when we measure them throughout the implementation set (or, dually, under interventions).**
>
> > …leftmost plot in Fig 2...misleading,...color coding here seems to refer to some threshold...rather than e.g. linearly representing the ambiguity as in the middle plot.
>
> The coloring represents the set of interpretations that we find to be ambiguous (see Lines 230-231). Here we are trying to emphasize that our metric is well calibrated in that it is selective enough to identify interpretations. Can the reviewer clarify what they mean by “linearly representing the ambiguity as middle plot?”
>
> > …(5.1), distance is defined between sets of variables where it seems that what is intended is some average over distances on actual inputs.
>
> Here, $d$ is essentially a function norm. For example, given some distribution $\mu$ over the inputs $X$, we can define $d$ to be $\int_S d(V(x), V’(x)) \mu(x)$. For generality, it is unimportant exactly what this norm is. We use this distance only through $d_\mathrm{repr}$ which must be Lipschitz continuous in this norm.
>
> > ...convinced of the importance of the theoretical component of this paper...elaborate more on the relation between the empirical and theoretical contributions?
>
> We developed the bulk of our theory first. Under the guidance of this theory, we then devised concrete algorithms and experiments. Let us first expand on the theory as this might make the connection clearer:
>
> 1.	The separate notions of a circuit, interpretation are not well-defined in the literature. Our first contribution is that we formalize the exact relationships between circuits, interpretations, and representations.
> 2.	This formalism allows us to define rigorously interpretive equivalence.
> 3.	In Main Result 1 and Main Result 2, we show that interpretive equivalence of two models depends crucially on one two quantities: (1) representation similarity of their implementations; (2) interpretive compression of the interpretation.
> At this point, the theory gives rise to Algorithm 1 through a couple of observations:
> 4.	Our key observation is that **both** quantities (representation similarity and interpretive compression) **can be estimated without access** to the model’s interpretation we just need a generating process for the implementations. **This corresponds to Algorithm 1 (which can be thought of as uniform MCMC sampling over implementations).**
> 5.	Then, Main Result 3 ties everything together in that it proves that the success of Algorithm 1 is lower bounded by interpretive equivalence.
>
> We concede that the organization of the paper doesn’t accentuate these connections. Our intention is to make practical portions of the paper accessible to those who make not be theoretically inclined and thus it should stand on its own. We welcome any organizational suggestions here.

---

> > ### Author Response · Authors · 2025-11-20
> >
> > > Detail of ambiguity figures…precise circuits being used here…ablations that are performed in ReprDist…
> >
> > Here, we use activation patching. To be concrete,
> >
> > 1.	We first identify a subset of the model’s attention heads that drive functional behavior. We do this through causal mediation analysis (which is well-documented in [Wang et al. (2023)](https://arxiv.org/abs/2211.00593) and [Tigges et al. (2024)](https://arxiv.org/abs/2407.10827)). For example, in GPT2-small this corresponds to attention heads (0.1, 0.10, 2.2, 4.11, 5.5, 6.9, etc., formatted as [layer index].[head index]). These attention heads form our set P (see Algorithm 1) and the complement of this set form N.
> > 2.	For any input like
> > ```
> > When John went to the store with Mary, John gave a bottle of milk to
> > ```
> > We construct a counterfactual input like
> > ```
> > When Mary went to the store with John, Mary gave a bottle of milk to
> > ```
> >
> > Then, we “Delete a subset of N uniformly” (Algorithm 1) by running the model on input (1) and then for attention head in the ablation subset we patch in the output of the attention head as if it had been run on (2).
> >
> > > [2]388: what does “yield the same interpretation” mean?
> >
> > Does the reviewer mean Line 388? There does not seem to be a reference to this quote on Line 288. Here, we mean that there exists some new alignment map (see Lines 357-360) which allows the interpretation of $K$ to also be an abstraction of $K$ (under the new weight configuration).
> >
> > > 156: we sample implementations, but according to what distribution? It seems in practice what we do is just what Algorithm 1 says?
> >
> > Our heuristic derivation in 156-160 holds for any distribution by symmetry. Also, yes, in practice we adhere to Algorithm 1. One way to reason about Algorithm 1 is that it is essentially an MCMC sampling procedure over the uniform distribution of implementations where Steps 2-6 are our proposal mechanisms and Step 7 is our acceptance
> >
> > We sincerely thank the reviews detailed feedback. We hope that we have addressed your concerns and look forward to further discussion with you.
> >
> > Best,
> >
> > Authors

---

### Official Review · Reviewer_K2kq · 2025-11-01

**Soundness:** 4
**Presentation:** 3
**Contribution:** 3
**Rating:** 6
**Confidence:** 3

**Summary:**

This work introduces the framework of interpretive equivalence (IE) in neural networks, which formalizes what it means for two neural networks to "do the same thing". The authors define notions of interpretation (high-level algorithmic process) and implementation (possible low-level realizations). Using the principle that models with non-distinguishable implementations must contain the same interpretation, the authors develop algorithms to check for interpretive equivalence, relying on the concept of ambiguity (are two implementations statistically different?). They validate their algorithms on several real-world models (Pythia, GPT-2) and tasks (permutation detection, IOI, POS/next token prediction) and ground their framework in the theoretical foundations of causal abstraction.

**Strengths:**

- This work focuses on an important and only recently acknowledged challenge in MI, which is the lack of rigor and solid theoretical foundations. The framework of interpretive equivalence bridges that gap by helping formalize what it means for two models to be equivalent in the algorithmic sense. The authors move away from most previous approaches that focus on a single interpretation (sometimes hard to define) and now focus on the much larger set of its implementations, which should make future automated discovery methods much more robust.
- The theoretical foundations were not checked down to the details, but they seem rigorous and strong. The authors build upon the framework of causal abstraction used in previous works and successfully formalize the concepts of interpretations, circuits and representations. In addition, the main theoretical results (1 and 2) provide necessary and sufficient bounds that connect IE to representation similarity, which is an important contribution. This brings much-needed mathematical precision to a field that often relies too heavily on empirical heuristics.
- The experimental validation is well-designed and compelling, progressing from toy tasks to real-world applications. The authors first demonstrate that their Ambiguity metric is well-calibrated using hard-coded RASP programs with known different interpretations on a toy task (permutation detection), then show the utility of their framework in comparing different models (GPT-2 and Pythia) on the IOI task. This confirms known results but shows the potential of IE to reduce the interpretation of large LLMs to smaller equivalent ones. Finally, the experiment in 3.3 relates next-token prediction (a complex task) to POS identification (which is much more well-understood), which shows that IE can be used to decompose complex problems into simpler ones.
- This is a rather complex work at the intersection of theory and practice, and effort has been put into making it rather accessible (presence of a glossary, high-level features).

**Weaknesses:**

- The success of the proposed Algorithm 1 seems to critically rely on the GetImpl procedure, which samples implementations from a model. This procedure is described at a very high level and could use more details. In particular, it is not clear how feasible and costly this procedure is for large LLMs. The quality of the Ambiguity score depends on how well these samples represent the actual set of all possible implementations. The paper does not discuss this limitation or the sensitivity of Ambiguity to the sampling process.
- The authors choose a metric based on linear transformability (drepr) for representation similarity. They mention that representations may however lie on complex and non-linear manifolds. This choice is common but could limit the framework's ability to detect more complex relationships. A discussion of how the framework would work with other types of similarity metrics would be useful.
- Ambiguity is not a binary score, but a graded one. It is not entirely clear how MI researchers should interpret these values. For example, what does an ambiguity score of 0.5 mean? Do the models share 50% of their algorithm? Considering the practical goal of the framework, the paper could include a discussion on how to choose proper thresholds or generally interpret these continuous scores.
- The introduced framework assumes that high representational similarity is evidence of interpretive equivalence, but other factors could lead to representational similarity (architectural biases, similarity in training data). The framework does not explicitly account for this.

**Questions:**

- How sensitive is Ambiguity to the number of implementations (n in Alg. 1) and the methods used to generate them (perturbations, deletions, etc.)? Have the authors explored how this scales with model size?
- How should one interpret graded values of the Ambiguity score?

---

> ### Author Response · Authors · 2025-11-20
>
> We sincerely appreciate the reviewer’s valuable time and feedback. To address your concerns, we have made several revisions to our submission and in addition we have also attached further clarifications to your questions and concerns below. We look forward to continued engagement with you during this discussion period.
>
> > The success of the proposed Algorithm 1 seems to critically rely on the GetImpl procedure, which samples implementations from a model. This procedure is described at a very high level and could use more details.
>
> We describe the operations of GetImpl in detail here:
>
> 1.	To find the sets of $N$ and $P$, the sets of components whose ablation preserves/degrades performance, respectively, we apply activation patching ([Heimersheim and Nanda 2024](https://arxiv.org/abs/2404.15255)).
> a.	All components (i.e. attention heads) that see a decrease in model performance when we patch their output with some nonsense input are added to $P$. Then, all remaining attention heads go into $N$.
> b.	A concrete example of this is illustrated in Figure 5 (Appendix E.5)
> 2.	Line 4: We apply a rotation to the activation of attention heads in P.
> 3.	Line 6: We delete a subset of $N$ by patching in activations that correspond to a counterfactual / nonsense input to the subset of attention heads of $N$ (in the context of the IOI task, we provide more detail about how to construct these counterfactuals; in all remaining cases, we simply sample a random token sequence as the counterfactual).
>
> We have added revisions in the submission to further clarify this, see Appendix E.4.
>
>
> > …feasible and costly this procedure is for large LLMs.
>
> In Section 3.2, “Reduction of Complex Models,” we apply this method for models up to size 2.8b. What are the models/sizes of models that you had in mind? Asymptotically, the computational cost of our procedure is comparable to component scoring complexity. Given any model
>
> 1.	We require $O(N)$ forward and backward passes to find the circuit of the model (where $N$ is the number of samples; this can be done through edge-attribution patching see [Hanna et al. 2023](https://arxiv.org/abs/2403.17806)).
> 2.	For any given implementation, getting its representations require $O(M)$ forward passes where M is the number of samples we wish to use to estimate representation similarity.
>
> Thus, the complexity of Algorithm 1 is $O(N + MK + \mathrm{poly}(M))$ where $K$ is the number of implementations we sample. The $\mathrm{poly}(M)$ factor comes from estimating representation similarity. Latency of Algorithm 1 can be offset because it is "embarassingly parallel."
>
> > The authors choose a metric based on linear transformability (drepr) for representation similarity. They mention that representations may however lie on complex and non-linear manifolds. This choice is common but could limit the framework's ability to detect more complex relationships. A discussion of how the framework would work with other types of similarity metrics would be useful.
>
>
> Algorithm 1 is not dependent on any specific notion of representation similarity. If $d_\mathrm{repr}$ has the properties of a pseudo-metric, our formalisms of Ambiguity still hold along with Main Result 3 and Appendix H.
>
> For our theory, we acknowledge this limitation and discuss more expressive notions of representation similarity in Lines 451-461. Here, we identify a tradeoff between the complexity of d_repr, the tightness of Main Result 1 and 2, as well as the tractability of Algorithm 1. Consider two regimes:
>
> 1.	$d_\mathrm{repr}$ **is very expressive**, that is $d_\mathrm{repr}(R_1,R_2) \to 0$ for almost all pairs $R_1, R_2$. By Main Result 1, the upper bound on $d_\mathrm{interp}$ tightens. However, by Main Result 2, the lower bound on $d_\mathrm{interp}$ loosens (one can see this by simply rearranging some of the terms in Main Result 2).
> 2.	$d_\mathrm{repr}$ **is not expressive** (i.e. just an affine transformation as we have defined), this means that for all $R_1, R_2, d_\mathrm{repr}(R_1, R_2)$ could be large. This implies that the upper bound on $d_\mathrm{interp}$ is loose, but its lower bound tightens.
>
> **What are the implications of this?** As we increase $d_\mathrm{repr}$'s expressiveness the rate at which the lower and upper bounds tighten depend strongly on the interpretive compression. Our intuition here is that there exists some non-trivial duality between interpretive compression (i.e. the geometry of the implementation space), the sample complexity of Algorithm 1, and $d_\mathrm{repr}$. Concretely, we hypothesize if the geometry of the implementation space is complex, then one needs make complex $d_\mathrm{repr}$ to achieve the same sample complexity in Algorithm 1.
>
> We think that making concrete claims about such interactions would require additional structural assumptions about the implementation space which is out of the scope of this paper, but would be interesting for future work!

---

> ### Author Response · Authors · 2025-11-20
>
> > Graded score of Ambiguity…
>
> We formally define Ambiguity in Appendix H. Concretely, if $M_1, M_2$ are two models, then it is the probability that representations $R_1, R_2 \sim \mathrm{ImplementationsOf}(M_1)$ no closer than representations $R_1 \sim \mathrm{ImplementationsOf}(M_1), R_2 \sim \mathrm{ImplementationsOf}(M_2)$. Main Result 3 gives some insight into graded notions of ambiguity. Ambiguity is an upper bound on the probability that two models’ representation similarity upperbounds interpretive equivalence. In other words, **ambiguity is proportional to how confident a model’s representations similarity upperbounds their interpretive equivalence.**
>
> > The introduced framework assumes that high representational similarity is evidence of interpretive equivalence, but other factors could lead to representational similarity (architectural biases, similarity in training data). The framework does not explicitly account for this.
>
> We do not assume that high representation similarity is evidence of interpretive equivalence. We actually prove this in Main Result 1, 2, and 3. **Our framework does account for the fact that representation similarity does not uniquely identify mechanistic interpretations. (see discussion in Section 5, Lines 380-418)** Notably, interpretive equivalence depends on both representation similarity and interpretive compression. **Interpretive compression accounts for potential non-identifiability of representations** (which is why Algorithm 1 seeks to enumerate entire sets of implementations rather than comparing models pointwise).
>
> > …sensitivity of Ambiguity to the sampling process…Have the authors explored how this scales with model size?
>
> We explore sensitivity of Ambiguity to model size in Section 3.2. Herein, we sample the same number of implementations (10) regardless of model size (from 160M to 2.8b). As shown in Figure 2(Center), we find that the efficacy of Algorithm 1 is not affected. Theoretically, we analyze the sample complexity of Algorithm 1 in Appendix H. We can bound this complexity log of the covering number of the implementation space.
>
> Thank you again for your thoughtful review! We look forward to hearing from you soon.
>
> Best,
>
> Authors

---

> > ### Comment · Reviewer_K2kq · 2025-11-24
> >
> > I thank the authors for their detailed and high-quality rebuttal.
> >
> > I appreciate the authors pointing me to the specific details in the paper and appendix. The clarification regarding the operational steps (activation patching, rotation, deletion) and the complexity analysis solves my concerns about the feasibility of the algorithm. The fact that this was demonstrated on 2.8B parameter models in the text confirms the scalability.
> >
> > The discussion regarding the trade-off between metric expressiveness and the tightness of the bounds in Main Results 1 & 2 is very helpful context. I appreciate the clarification that the "Interpretive Compression" part of the framework (and the associated main results) takes into account the potential disconnect between representation similarity and interpretation.
> >
> > The rebuttal has resolved my main questions and highlighted where the necessary theoretical and practical details are located in the manuscript. The paper appears to be rigorous and the experimental validation is sound. I therefore maintain my positive rating.

---

### Author Response · Authors · 2025-11-20

We sincerely thank all reviewers for their time and high-quality feedback. We hope that you will find our clarifications and response to your individual comments helpful. At this time, we incorporated your feedback through several revisions in the updated PDF (fixes are marked in blue and new content is marked with red). We summarize them here:

1.	We fixed several typos throughout the manuscript that were identified by Reviewer `DWxu`.
2.	Following feedback from all reviewers, we have added a new section in the appendix (Appendix A: Questions and Clarifications) that address your concern as potential future readers of the paper may also share them.
3.	We have additionally clarified the IOI experimental setting along with GetRepr / GetImpl in Appendix E.4.

We hope that these modifications may increase your confidence in our submission, and we look forward to further engaging with you during this discussion period!

---

### Author Response · Authors · 2025-12-03
**Summary of Reviewers+Rebuttal (1/2)**

We sincerely appreciate the valuable feedback and engagement from all reviewers during the discussion period. Your feedback has improved the manuscript and we were looking forward to address more of your questions and potential concerns. At this point, we would like to offer a summary of our interactions.

## Summary of Paper
- We introduce a **formal definition of interpretive equivalence**: a criterion for when two neural networks share the “same interpretation” without having to specify *what* their interpretations are.
- We provide **tractable algorithms** to estimate interpretive equivalence. This comes with **theoretical guarantees**: necessary and sufficient conditions linking circuit equivalence, representation similarity, and interpretive equivalence. To our knowledge, this is the first time a unified theory between all of these objectives connection is established.
- We **empirically demonstrate** that interpretive equivalence enables "reductions." For example, we can interpret a large model by studying a smaller one that is interpretive equivalent. And, decompose a model's behavior on a complex task on simpler, interpretive equivalent ones.
	- Experiments on 2.8B-parameter models show our method works beyond toy examples, and synthetic benchmarks (e.g., based on programmatic tasks) validate the ability of our metric (called **Ambiguity**) to distinguish qualitatively different interpretations.

## Summary of Reviewers' Strengths
- **Novel, well-motivated problem setting** (Reviewer pr5o, K2kq, Td1h):
	- "The shift from 'interpret each model' to 'determine if models share interpretations' is valuable and practical as a conceptual idea";
	- "[this] should make future automated discovery methods much more robust"
	- "The idea of quantifying representational similarity across a circuit...is an interesting approach that sidesteps some of the intractable parts of MI"
- **Rigorous theoretical foundations** (Reviewer Td1h, DWxu, K2kq):
	- "The paper makes a serious attempt to ground the somewhat informal terms used in MI in the precise language of casual graphs";
	- "nice mix of theory and practice...";
	- "brings much-needed mathematical precision to a field that often relies too heavily on empirical heuristics."
- **Clear presentation of theory** (Reviewer K2kq, Td1h, pr5o, DWxu):
	- "rather complex work at the intersection of theory and practice, and effort has been put into making it rather accessible";
	- "significant amount of new terminology in this paper, but the authors do provide a detailed glossary which I appreciated";
	- "I appreciate that the authors put effort in the presentation of the work and the concepts."
	- "paper is well written and its connection to previous work is evidently thought-through throughout the paper."
- **Strong experimental design** (Reviewer K2kq, Td1h):
	- "experimental validation is well-designed and compelling, progressing from toy tasks to real-world applications"
	- "show[s] the feasibility of applying the method at scale"
	- "evidence...that this framework can distinguish between implementations on the toy task...is promising"

---

> ### Author Response · Authors · 2025-12-03
> **Summary of Reviews+Rebuttal (2/2)**
>
> ## Summary of Reviewers' Concerns & How We Addressed Them
>
> > **Feasibility/Scalability/Complexity**
>
> We clarified the steps of `GetRepr` and `GetImpl` in the revised version of the manuscript. We pointed reviewers to sample complexity analyses in Appendix H. And also, emphasized that our experiments on the  2.8B Pythia model support that our method scales. Reviewer K2kq explicitly that these clarifications addressed their concerns.
>
> > **Representational Similarity vs. Interpretive Equivalence**: Does representational similarity necessarily imply interpretive equivalence? Are our proposed algorithms sufficiently discriminative?
>
> We emphasized (and proved) that under our formal definition, interpretive equivalence does _not_ collapse to naive representational similarity. We clarified how "interpretive compression”  account for possible disconnects between representation and interpretation. Both Reviewer DWxu and K2kq acknowledged this as “very helpful.” Reviewer pr5o also agrees with this conceptually, but disagrees with the empirical results (see below).
>
> > **Empirical scope/generality**: Are the experiments convincing beyond toy settings?
>
> We want to emphasize that RASP programs to benchmark mechanistic interpretability algorithms is standard ([Conmy et al., 2023](https://arxiv.org/abs/2304.14997); [Friedman, Wettig, and Chen, 2023](https://arxiv.org/abs/2306.01128); [Lindner et al., 2023](https://arxiv.org/abs/2301.05062); [Thurnherr and Scheurer, 2024](https://arxiv.org/abs/2409.13714); [Mondorf, Wold, and Plank, 2025](https://aclanthology.org/2025.acl-long.727.pdf); [Yu et al., 2025](https://arxiv.org/pdf/2407.03779)). This is because RASP programs are easy to understand and evaluate, i.e. most people can agree when two models point to the same RASP programs. Further, we want to emphasize here that even though we are using RASP programs, we are **not** using RASP transformers (the method of [Gupta et al., 2024](https://arxiv.org/abs/2407.14494) generates models with weight distribution almost identical to as if they have been trained using SGD). For each RASP program, we are also generating implementations with many different architectures.
>
> Importantly, reviewers recognized the value of our contribution despite that limitation:
> - Reviewer pr5o wrote they “would be completely satisfied to see this paper published at ICLR.”
> - Moreover, Reviewer DWxu wrote "[this paper] will be a nice contribution to the conference"
> - Reviewer K2kq wrote "the paper appears to be rigorous and the experimental validation is sound."
>
> Thank you again to all reviewers that took the time to engage with our work and rebuttal. We value your feedback and effort. We are happy that the paper is now more clear and polished as a result of this back-and-forth. We hope that this summary will be helpful to the new Area Chair and thank all of you for your time.
>
> Best,
>
> Authors

---

### Meta-Review · Area_Chair_oBE5 · 2026-01-07

**Summary:**

This paper proposes a framework to determine interpretive equivalence between neural networks. The key idea is to solve a relaxed problem where two interpretations are equivalent if and only if the set of possible implementations are similar. Then the authors develop algorithms to check for interpretive equivalence and validate the algorithms on several models and tasks. Overall, most reviewers find the contributions of this paper useful, especially on the construction of theoretical framework.

While Reviewer pr5o's additional comments are not addressed (concerns on Fig 2, lacking empirical demonstrations, and narrow scope synthetic experiments), most reviewers concerns are addressed. Hence an acceptance is recommend. The authors are urged to address reviewer pr5o's concerns in the camera-ready version.

**Reviewer Concerns:**

* Reviewer K2kq's concerns are addressed and stated to remain positive rating
* Reviewer Td1h's concern on insufficient related work, importance of the theoretical components of this paper, and deatils of Fig 2 are addressed
* Reviewer pr5o's concern on Fig 2, empirical demonstration on counter examples, narrow scope of synthetic experiments, and no practical robustness of AMBIGUITY are outstanding
* Reviewer DWxu's concerns are addressed

**Reviewer Scores:**

* Reviewer K2kq stated to remain positive rating of 6
* Reviewer Td1h likely increase score to 5
* Reviewer pr5o stated to remain score of 6 but think it'll be better to address additional concerns
* Reviewer DWxu stated to remain high score of 8

---

### Decision · Program_Chairs · 2026-01-26

Accept (Poster)